# Photocatalytic and Cathode Active Abilities of Ni-Substituted α-FeOOH Nanoparticles

**DOI:** 10.3390/ijms241814300

**Published:** 2023-09-19

**Authors:** Ahmed Ibrahim, Mikan Shiraishi, Zoltán Homonnay, Stjepko Krehula, Marijan Marciuš, Arijeta Bafti, Luka Pavić, Shiro Kubuki

**Affiliations:** 1Department of Chemistry, Graduate School of Science, Tokyo Metropolitan University, Tokyo 192-0397, Japan; elawadyahmed59@gmail.com (A.I.); shiraishi-mikan@ed.tmu.ac.jp (M.S.); 2Institute of Chemistry, Eötvos Loránd University, 1117 Budapest, Hungary; homonnay.zoltan@ttk.elte.hu; 3Division of Materials Chemistry, Ruđer Bošković Institute, 10000 Zagreb, Croatia; stjepko.krehula@irb.hr (S.K.); marijan.marcius@irb.hr (M.M.); lpavic@irb.hr (L.P.); 4Faculty of Chemical Engineering and Technology, University of Zagreb, Marulićev trg 20, 10000 Zagreb, Croatia; abafti@fkit.hr

**Keywords:** goethite nanoparticles, impedance spectroscopy, Fourier transform infrared spectroscopy (FT-IR), visible-light-activated photocatalyst, cathode active material, sodium-ion batteries, lithium-ion batteries

## Abstract

The present study investigates the relationship between the local structure, photocatalytic ability, and cathode performances in sodium-ion batteries (SIBs) and lithium-ion batteries (LIBs) using Ni-substituted goethite nanoparticles (Ni_x_Fe_1−x_OOH NPs) with a range of ‘x’ values from 0 to 0.5. The structural characterization was performed applying various techniques, including X-ray diffractometry (XRD); thermogravimetry differential thermal analysis (TG-DTA); Fourier transform infrared spectroscopy (FT-IR); X-ray absorption spectroscopy (XANES/EXAFS), both measured at room temperature (RT); ^57^Fe Mössbauer spectroscopy recorded at RT and low temperatures (LT) from 20 K to 300 K; Brunauer–Emmett–Teller surface area measurement (BET), and diffuse reflectance spectroscopy (DRS). In addition, the electrical properties of Ni_x_Fe_1−x_OOH NPs were evaluated by solid-state impedance spectroscopy (SS-IS). XRD showed the presence of goethite as the only crystalline phase in prepared samples with x ≤ 0.20, and goethite and α-Ni(OH)_2_ in the samples with x > 0.20. The sample with x = 0.10 (Ni10) showed the highest photo-Fenton ability with a first-order rate constant value (*k*) of 15.8 × 10^−3^ min^−1^. The ^57^Fe Mössbauer spectrum of Ni0, measured at RT, displayed a sextet corresponding to goethite, with an isomer shift (*δ*) of 0.36 mm s^−1^ and a hyperfine magnetic distribution (*B*_hf_) of 32.95 T. Moreover, the DC conductivity decreased from 5.52 × 10^−10^ to 5.30 × 10^−12^ (Ω cm)^–1^ with ‘x’ increasing from 0.10 to 0.50. Ni20 showed the highest initial discharge capacity of 223 mAh g^−1^, attributed to its largest specific surface area of 174.0 m^2^ g^−1^. In conclusion, Ni_x_Fe_1−x_OOH NPs can be effectively utilized as visible-light-activated catalysts and active cathode materials in secondary batteries.

## 1. Introduction

Metal oxide nanoparticles (MO NPs) exhibit a large surface area and a pronounced quantum size effect due to superparamagnetism and the quantum tunnelling phenomenon, distinguishing them from bulk MO materials. Among the wide array of MO NPs, iron oxide NPs have emerged as a novel functional material class that has gained considerable attention across various domains. These include biomedical applications, wherein they demonstrate antimicrobial and anticancer activities [1], as well as catalysts [2], photocatalysts [3,4], and agents for environmental purification [5].

In particular, goethite (α-FeOOH) NPs represent the most readily available form of iron oxyhydroxide in nature. Their crystal structure comprises interconnected FeO_3_(OH)_3_ octahedra with shared edge and vertex linkages. The versatile applications of α-FeOOH NPs encompass traditional roles as pigments, catalysts, photocatalysts [6], adsorbents, gas sensors, photoelectrodes [7], and battery electrodes [8,9]. Notably, the utilization of α-FeOOH NPs for wastewater purification and as secondary battery electrodes holds immense significance in realizing future sustainable development goals. This is primarily due to their ability, non-toxic nature, stability, electrical conductivity, and photocatalytic activity under visible-light exposure [10].

As for the UV- and visible-light-activated photocatalytic ability of α-FeOOH NPs, a first-order rate constant (*k*) of 7.4 × 10^−4^ min^−1^ was recorded for 15 mg of hematite (α-Fe_2_O_3_) NPs formed by the combustion method, in a reaction with 50 mL of an aqueous solution containing 10 mg of Rhodamine B (RhB) under UV light irradiation [3]. In this reaction, 95% of the RhB decomposed after 180 min [3]. In addition, the photocatalytic reaction between RhB–H_2_O_2_ aqueous solutions and 10 mg of powdered, 4 mol% Cu-doped goethites under exposure to visible light resulted in a 60% degradation of RhB in 1 h, which could be attributed to the combined effect of nanoneedle morphology and Cu doping [11].

In relation to the developments of visible-light-activated photocatalysts and secondary battery electrodes of iron oxide NPs, we confirmed the methylene blue (MB) degradation effect of a mixture of Fe NPs and γ-Fe_2_O_3_ NPs [12]. In this study, a 10-day MB_aq_ degradation test using a 1:3 molar ratio of Fe NPs to γ-Fe_2_O_3_ NPs mixed powder showed a significant decrease in MB concentration from 20.0 to 0.85 μM with a maximum *k* value of 2.84 day^−1^, indicating that iron oxide NPs effectively degrade organic compounds [13]. This result proves that the nano size of Fe and γ-Fe_2_O_3_ showed larger reactivity because of their high surface area. Another result of methylene blue decomposition was reported on Sn-incorporated FeOOH NPs, showing a smaller band gap energy and higher decomposition performance compared to pure FeOOH. It is assumed that the band gap energy of α-FeOOH NPs decreased due to the introduction of Sn^IV^ [14].

On the other hand, as for the application of α-FeOOH as an electrode for a secondary battery, an excellent cycling performance with a reversible specific capacity of 870 mAh g^−1^ under the current rate of 100 mA g^−1^ of LIB was reported. The anode contained α-FeOOH corner-truncated prisms (CTPs) with a length of approximately 1 μm and a width of approximately 200 nm obtained by hydrothermal treatment [9].

An intriguing and notable characteristic of α-FeOOH NPs is their inherent capability to incorporate diverse metal cations into their crystal structure. This opens the field to the possibility of modifying various crystal properties, including morphology, size, crystallinity, and colour, by replacing Fe^3+^ ions within the structure. In particular, the doping of α-FeOOH NPs with a range of metal cations, such as Al^3+^, In^3+^, Cu^2+^, Cr^2+^, Mn^2+^, and Zn^2+^, was extensively investigated to enhance further the aforementioned physical properties [11,12]. These studies have explored the potential for improving and refining the characteristics of α-FeOOH NPs by introducing and incorporating these selected metal cations into their structure.

In addition, Cu^2+^-for-Fe^3+^ substitution in goethite caused a gradual elongation and narrowing of nanorods with the formation of nanoneedles, which led to a slight decrease in direct and indirect optical band gaps compared with the pure phases [11]. The introduction of nickel (Ni) as a dopant in α-FeOOH NPs is anticipated to impact their crystalline structure and oxidation state. Ni can substitute for Fe ions within the crystal lattice of α-FeOOH NPs [12]. This incorporation of Ni is likely to result in slight modifications of the crystalline structure, which predominantly consists of interconnected FeO_6_ octahedra with shared edges and vertices [11,12]. Upon doping with Ni, the dopant ions can occupy available sites within the lattice [11,12]. The extent of the structural changes depends on many factors such as the concentration of the dopant and its compatibility with the lattice structure of α-FeOOH [15].

The oxidation state of Ni within the doped α-FeOOH NPs primarily depends on the specific synthesis conditions and the prevailing redox environment [15]. Typically, Ni is commonly found in a +2-valence state. However, it is important to note that the local chemical environment and interactions with neighbouring atoms within the lattice structure can influence the oxidation state of the dopant ions.

Herein, we report the relationship between the structure, photocatalytic ability, and active cathode property of a secondary battery of Ni-substituted α-FeOOH NPs prepared by hydrothermal synthesis. For this purpose, structural characterization was performed by X-ray diffractometry (XRD), X-ray absorption spectroscopy (XANES/EXAFS), and ^57^Fe-Mössbaur spectroscopy. Moreover, for the properties of photocatalytic ability, electrochemical performance, and active cathode evaluation, Brunauer–Emmett–Teller (BET) surface area analysis, diffuse reflectance ultraviolet-visible (UV-Vis) spectroscopy, impedance spectroscopy (IS), and active cathode performance of LIBs and SIBs were carried out.

## 2. Results and Discussion

### 2.1. XRD Patterns of α-Ni_x_Fe_1−x_OOH NPs

Figure 1 shows the XRD patterns of Ni0, Ni5, Ni10, Ni15, Ni20, Ni30, Ni40, and Ni50 samples. The XRD peaks of goethite (ICDD card No. 00-029-0713) with the space group (Pbnm) were observed in all XRD patterns with increasing Ni content. Moreover, a new peak appeared at 2*θ* = 11.56° corresponding to the (0 0 3) plane of α-Ni(OH)_2_ crystalline phase (ICDD card No.01-076-6904) with the space group (R3m: H), and the intensity of this peak increased with the increasing x content from 0.30 to 0.50, as shown in the yellow rectangle in Figure 1. The percentage of the α-Ni(OH)_2_ crystalline phase increased from 7.63% to 9.35% while ‘x’ increased from 0.30 to 0.50, as shown in Table 1. Additionally, the elevation of background intensity observed in the XRD patterns depicted in Figure 1, as the Ni content increased from 0.30 to 0.50, suggests the appearance of an amorphous Ni hydroxide phase. The diffraction lines of α-FeOOH were slightly shifted towards lower 2*θ* values with the increasing Ni content, implying a minor expansion in the unit cell due to Ni/Fe substitution. The unit cell parameters of goethite in the Ni0, Ni5, Ni10, Ni15, Ni20, Ni30, Ni40, and Ni50 samples obtained by Rietveld refinement of the XRD patterns are listed in Table 1. The slight variation in ionic radius between Fe^3+^ (0.645 Å) and Ni^2+^ (0.69 Å) [16] ions caused a small expansion of the unit cell of α-FeOOH. In line with the previous studies [17,18], it was observed that the expansion along the *b*-axis direction was most remarkable. The non-uniform expansion is likely to be attributed to different distortions in the octahedral sites occupied by Ni^2+^ and Fe^3+^ ions, stemming from their disparate electron configurations (3d^8^ and 3d^5^, respectively). The incorporation of Ni into α-FeOOH can be attributed partially to its incongruent release from the initial Fe-Ni hydroxide (Ni-ferrihydrite) [19]. The calculated average size of α-Ni_x_Fe_1−x_OOH NPs by the Scherrer equation from the FWHM of the (110) diffraction line is summarized in Table 1.

### 2.2. Thermogravimetric and Differential Thermal Analysis (TG-DTA) of α-Ni_x_Fe_1−x_OOH

Thermogravimetric (TG) and differential thermal analysis (DTA) curves for Ni_x_Fe_1−x_OOH NPs with ‘x’ 0, 0.10, 0.20, 0.30, 0.40, and 0.50 are shown in Figure 2a,b, respectively. The weight loss at temperatures up to approximately 140 °C corresponds to the release of H_2_O molecules adsorbed on the surface of Ni_x_Fe_1−x_OOH NPs crystals or trapped within the interstitial spaces between them [19,20]. This low-temperature weight loss increased in the Ni_x_Fe_1−x_OOH NPs samples with the increased Ni content, which can be explained by more adsorbed water in the NiFeOOH NPs samples with a large surface area [20]. Furthermore, different molar fractions of incorporated Ni^2+^ ions and different crystal facets in Ni-doped Ni_x_Fe_1−x_OOH NPs crystals can also affect the amount of adsorbed water in Ni_x_Fe_1−x_OOH NPs samples. The weight loss at temperatures between approximately 150 and 290 °C can be attributed to the dehydroxylation of the surface hydroxyl groups [18,19,20]. This weight loss also increases in the Ni_x_Fe_1−x_OOH NPs with a higher Ni content due to the larger surface area. The total weight loss percentage for Ni_x_Fe_1−x_OOH NPs in the temperature range from RT to 1000 °C increased when the Ni content increased, from 14.24% in Ni0 to 26.63% in Ni50, as shown in Figure 2a.

Figure 2b shows the DTA curves of the Ni_x_Fe_1−x_OOH NPs with ‘x’ 0, 0.10, 0.20, 0.30, 0.40, and 0.50. The DTA curve of Ni_x_Fe_1−x_OOH NPs showed a single endothermic peak in the temperature range from 230–300 °C due to the phase transformation from α-FeOOH to α-Fe_2_O_3_ [18,19]. As the amount of Ni increased from 0 to 20 mol%, the endothermic peak became smaller and somewhat shifted to higher temperatures from 251 °C in Ni0 to 262 °C in Ni20, while in Ni30, it shifted in the opposite direction and was observed at 241 °C. The intensity of the endothermic peak of Ni30 was highly diminished compared to the rest of the samples. It seems to be related to the appearance of a second minor α-Ni(OH)_2_ crystalline phase in Ni30, see Figure 1. As the Ni increases, a new endothermic peak appears in Ni40 and Ni50 at lower temperatures 150–200 °C, caused by the lower degree of crystallinity or the increased amorphous phase [19], in addition to the resolved second one at 250–260 °C. These results agree with the XRD data (see Figure 1 and Table 1).

### 2.3. FTIR Spectra of α-Ni_x_Fe_1−x_OOH NPs

The FTIR spectra of Ni_x_Fe_1−x_OOH NPs with ‘x’ 0, 0.10, 0.20, 0.30, 0.40, and 0.50, and Fe_2_O_3_ and NiO as standard reference materials, carried out in the wave number range of 400–4000 cm^−1^, are shown in Figure 3. The transmission band showed approximately 3454 cm^−1^ ascribed to -OH [21] in Ni0 and shifted to a lower wave number, and the peak intensity increased with the increasing Ni content. The small transmission peak observed at 3130 cm^−1^ corresponds to the O-H stretching vibration of the hydroxyl group [18,22]. The band centre of the hydroxyl group gradually shifts to a higher wave number, and the peak intensity decreases with the increasing Ni content, as shown in Figure 3. The intense two transmission peaks were observed at 797 cm^−1^ and 895 cm^−1^ related to the Fe-O-H bending vibrations out of plane and in plane [17,18,19,22], respectively. The intensity of both transmission peaks gradually decreased with the Ni content. These two bands slightly shifted towards higher wave numbers with up to 30 mol% Ni content, and with high Ni content (40 and 50 mol%), the centre of these bands changed to lower wave numbers. It is known that the bending bands’ positions depend on the crystallinity degree and the transition metal substituted with Fe [17,22]. In this study, Ni substituted with Fe caused a shift toward lower wave numbers with the increasing Ni content from Ni0 to Ni30, while in Ni40 and Ni50, an increase in particle size caused a change to a higher wave number. These results agree with XRD and DTA (see Figure 1 and Figure 2). The transmission band was observed at 645 cm^−1^ in pure-FeOOH due to Fe-O symmetric stretching [22], and the band centre gradually shifted to a lower wave number with the increasing Ni content from 0 to 0.5. A new transmission peak appeared in higher Ni concentration samples (Ni40 and Ni50) at 472 and 476 cm^−1^, respectively. It was noted that the recent peak compared with that shown in the FTIR spectrum of NiO can be attributed to Ni-O [23]. The small transmission bands located at 1639 cm^−1^ in Fe_2_O_3_ are due to the stretching vibration of -OH from the absorbed water [21,23,24]. Furthermore, this peak showed in Ni_x_Fe_1−x_OOH NPs at 1639 cm^−1^ in Ni0 and slightly shifted to 1643 cm^−1^, and the peak intensity gradually increased with the increasing Ni content.

### 2.4. XAFS Spectra of α-Ni_x_Fe_1−x_OOH NPs

The XAFS and EXAFS measurements are powerful tools for investigating the local atomic and electronic properties of materials, providing a comprehensive understanding of their structure and reactivity [25,26].

Figure 4 shows the XANES spectra and Fourier transforms of Fe-/Ni-*K*-edges of α-Ni_x_Fe_1−x_OOH NPs with ‘x’ 0, 0.05, 0.10, 0.15, 0.20, 0.30, 0.40, and 0.50 and Fe foil, Fe_3_O_4_, and α-Fe_2_O_3_ as standard materials in the Fe *K*-edge and with ‘x’ from 0.10, 0.15, 0.20, 0.30, 0.40, and 0.50 and Ni-foil and NiO as reference materials at the Ni *K*-edge.

Figure 4a displays the normalized Fe *K*-edge XAFS spectra of α-Ni_x_Fe_1−x_OOH NPs. The Fe *K* absorption edge was observed at a normalized intensity = 0.5 for Fe-foil, Fe_3_O_4_, and Fe_2_O_3_ being 7117.26 ± 0.02, 7120.67 ± 0.02, and 7121.79 ± 0.02 eV, respectively. The oxidation state in Fe-foil, Fe_3_O_4_, and α-Fe_2_O_3_ is 0, 2.66, and 3, respectively. The absorption edge of α-Ni_x_Fe_1−x_OOH NPs was observed at 7122.65 ± 0.01, 7122.49 ± 0.01, 7122.27 ± 0.01, 7122.15 ± 0.01, 7122.23 ± 0.01, 7121.91 ± 0.01, 7122.08 ± 0.01, and 7122.28 ± 0.01 with ‘x’ 0, 0.05, 0.10, 0.15, 0.20, 0.30, 0.40, and 0.50, respectively, as shown in the insert figure in Figure 4a. The increasing oxidation state of Fe results in a shift of the absorption edge to higher energy [27]; according to the above, the values of the absorption edge of α-Ni_x_Fe_1−x_OOH NPs are higher than the values of Fe_2_O_3_ (Fe^3+^), this implies that none of the α-Ni_x_Fe_1−x_OOH NPs samples contained Fe^2+^, and they contained only Fe^3+^. The pre-edge peak was found at 7113 eV for α-Ni_x_Fe_1−x_OOH NPs and at 7114 eV for both Fe-foil and Fe_2_O_3_, being designated as the absorption edge. The pre-edge features in the XANES spectrum reveal electronic states and symmetry of the unoccupied orbitals in the absorbing atom [27,28]. Analyzing the intensity, energy position, and shape of pre-edge peaks helps understand the electronic structure, oxidation state, and local environment of Fe atoms in the material [29]. The intensity of the pre-edge peak is lower when the octahedral site has high symmetry. α-FeOOH with the octahedral structure of FeO_6_ shows an increase in the pre-edge peak intensity and, therefore, a decrease in symmetry with the increasing Ni content.

Figure 4c displays the normalized Ni *K*-edge XANES spectra of Ni_x_Fe_1−x_OOH NPs with ‘x’ 0.10, 0.15, 0.20, 0.30, 0.40, and 0.50 and Ni-foil and NiO as reference materials. The pre-edge peak of Ni foils, NiO, and Ni_x_Fe_1−x_OOH NPs was shown at 8321.40, 8319.68, and 8319.48 eV, respectively [30]. The height of the pre-edge peak intensity of NiO and Ni_x_Fe_1−x_OOH NPs has approximately the same value, and the change in the pre-edge is minimal. Therefore, we cannot find any significant difference in the oxidation state of Ni and the symmetry of the structure [31].

In contrast, the Ni-*K* normalized absorption peak of Ni_x_Fe_1−x_OOH NPs and reference materials are shown in the inserted figure in Figure 4c. The absorption edge of Ni-foil, Ni_x_Fe_1−x_OOH NPs, and NiO is at 8327.40, (8329.21–8329.48), and 8328.76 eV, respectively. According to the absorption edge’s higher value compared to NiO, the oxidation state of Ni in all Ni_x_Fe_1−x_OOH NPs samples is higher than +2. Both Ni15 and Ni20 are samples containing less Ni^3+^, while Ni10 has the highest level of Ni^3+^.

Figure 4b shows the Fourier transform of Fe-*K*-edge EXAFS (FT-EXAFS) for α-Ni_x_Fe_1−x_OOH NPs. The FT-EXAFS of α-Ni_x_Fe_1−x_OOH NPs, in the first coordination, was observed at 1.57 Å, and the peak intensity decreased with the increasing ‘x’ from 0.10 to 0.50, and this peak could be attributed to Fe-O [29,30]. Moreover, the second peak of Fe-foil, Fe_3_O_4_, and Fe_2_O_3_ was shown at 2.21 Å, 2.91 Å, and 2.68 Å, respectively. The second peak of Ni_x_Fe_1−x_OOH NPs was also shown between the Fe_2_O_3_ and Fe_3_O_4_. Furthermore, this peak was shifted to a higher Fe-Fe distance, and the peak intensity decreased by reducing the Fe content from 1 to 0.50. All observed peaks in this range (2.21–2.91) Å are attributed to Fe-Fe (or Fe-Ni) [32,33].

Figure 4d shows the Ni-*K* FT-EXAFS of Ni_x_Fe_1−x_OOH NPs, and it was clear that the two peaks at 1.63 Å and 2.77 Å can be attributed to Ni-O and Ni-Me (Me = Ni or Fe), respectively [33,34,35]. The first peak was moved to a higher distance between Ni and O 1.75 Å, but the second peak shifted to a lower Ni-Me distance of 2.62 Å for Ni_x_Fe_1−x_OOH NPs [35]. The intensity of the first peak decreased, and the peak position was moved to a higher radius with the increasing Ni content. These results obtained from Ni-*K* and Fe-*K* FT-EXAFS and XANES of the Ni_x_Fe_1−x_OOH NPs agree with the XRD results.

### 2.5. ^57^Fe-Mössbauer Spectra of α-Ni_x_Fe_1−x_OOH NPs

Figure 5 shows the ^57^Fe-Mössbauer spectra of α-Ni_x_Fe_1−x_OOH NPs ‘x’ from 0.00 to 0.50 measured at RT in the velocity range from 12 mm s^−1^ to −12 mm s^−1^ and from 3 mm s^−1^ to −3 mm s^−1^. The Mössbauer parameters in these two velocity ranges are listed in Table 2 and Table 3. Figure 5a shows the ^57^Fe-Mössbauer spectra of Ni0, Ni10, Ni20, Ni30, Ni40, and Ni50 measured at RT in the wide velocity range from 12 mm s^−1^ to −12 mm s^−1^. The Mössbauer spectra exhibit a sextet with Mössbauer parameters characteristic of α-Ni_x_Fe_1−x_OOH NPs ‘x’ from 0.00 to 0.30 as the predominant component, while the sextet disappeared after increasing the Ni content to 0.30. The α-FeOOH NPs and α-Ni_x_Fe_1−x_OOH NPs spectra were analyzed by incorporating the hyperfine magnetic distribution (*B*_hf_); their average values are presented in Table 2. The slight changes in isomer shift, except for the Ni0 sample, indicate that the site is distorted goethite, as shown in Table 2. Moreover, the presence of a quadrupole doublet, which increases as the Ni content increases, can be ascribed to the presence of Fe^3+^ ions in a low-crystalline or amorphous phase. This phase could be attributed to compounds such as Ni-Fe hydroxide, low crystalline Ni-ferrite, or Ni-goethite. The higher proportion of the low-crystalline or amorphous phase aligns with the increased background observed in the XRD patterns, as depicted in Figure 1. Moreover, the line width value increased from 0.196 to 0.53 mm s^−1^, possibly confirming the result obtained from XRD, wherein the amorphous phases increased with the increasing Ni content. In addition, the particle size of the goethite drops to a nano-scale, and the superparamagnetic component increases as a function of the increasing Ni content [11,14,15]. The presence of the doublet suggests nanosized goethite particles in Ni40 and Ni50. To analyze the doublet component of the Mössbauer spectra of Ni20, Ni30, Ni40, and Ni50 measured at RT in the wide velocity range for more detail, a narrower velocity range from 3 mm s^−1^ to −3 mm s^−1^ was applied, and the spectra are presented in Figure 5b. One doublet shown in the Ni20, Ni30, Ni40, and Ni50 spectra in the wide velocity range from 12 mm s^−1^ to −12 mm s^−1^ was analyzed as a doublet component. The two doublets were considered to be derived from the superparamagnetic and amorphous (or ferrihydrite) components [15].

On the other hand, the Mössbauer spectra of Ni0, Ni5, Ni10, and Ni20 were measured in the range of velocity from 12 mm s^−1^ to −12 mm s^−1^ at 86 K, as shown in Figure 6a. Goethite was present in all spectra, and a doublet derived from amorphous (or ferrihydrite) is also present in Ni10 and Ni20. Additionally, the Mössbauer spectra of Ni20 and Ni50 were measured in a narrower range, as shown in Figure 6b and Appendix A.

A sextet of FeOOH and a doublet of amorphous (ferrihydrite) were present in Ni20, and Ni50 had two doublets derived from ferrihydrite and a superparamagnetic phase. These results are in good agreement with the XRD and XAFS measurements, as shown in Figure 1 and Figure 4. From Appendix A, the quadrupole splitting of the doublet derived from amorphous or ferrihydrite increased from 0.70 mm s^−1^ for Ni20 to 0.89 mm s^−1^ for Ni50. Therefore, it is considered that amorphous ageing is progressing, and ferrihydrite is being formed [17,19]. This is consistent with the fact that Ni50 has a specific surface area lower than Ni20, as was shown by BET analysis [35]. In addition, sample Ni20 contains goethite and an amorphous phase, while sample Ni50 also contains crystalline α-Ni(OH)_2_, which is a possible reason for the lower surface area. The Mössbauer parameter spectra of Ni10 and Ni20 were measured at low temperatures from 20 K to 300 K, as shown in Appendix A, and the Mössbauer parameters of Ni10 and Ni20 are summarized in Appendix A.

As shown in Table 2, the Mössbauer parameters measured at room temperature indicated a sextet and a doublet related to goethite and the superparamagnetic phase. Furthermore, the doublet shown at RT is divided into two doublets related to the superparamagnetic and amorphous phases. Additionally, with the decreasing temperature from 300 K to 20 K, the doublets disappeared in both samples Ni10 and Ni20.

### 2.6. TEM Observation of α-Ni_x_Fe_1−x_OOH NPs

The structural details revealed through X-ray diffraction (XRD) patterns, ^57^Fe-Mössbauer spectroscopy, as well as the X-ray absorption fine structure (XAFS) spectra of α-Ni_x_Fe_1−x_OOH NPs, closely correspond to the morphology and porosity evaluated via transmission electron microscopy (TEM) observations and the BET and BJH measurements. The average length, width, and ratio (length/width) of α-Ni_x_Fe_1−x_OOH NPs are evaluated from the TEM images in Figure 7, and the results are listed in Table 4. In addition, the histograms of particle length are shown in Figure 8. It was found that the average length (*c*-axis direction), average width (*b*-axis direction), and average ratio (length/width) of Ni0 are 181 ± 83 nm, 21.6 ± 6.6 nm, and 9.6 ± 6.5 nm, respectively [14].

From Figure 8a, it was found that the average length (*c*-axis direction), average width (*b*-axis direction), and average ratio (length/width) of Ni10 are 141 ± 54 nm, 33.2 ± 11.2 nm, and 4.6 ± 2.0 nm.

From Figure 8b, it was found that the average length (*c*-axis direction), average width (*b*-axis direction), and average ratio (length/width) of Ni20 are 263 ± 179 nm, 42.9 ± 13.9 nm, and 6.3 ± 3.6 nm. Ni10 forms nanorods that are shorter in length and wider in width than Ni0. Ni20 has the longest length and widest width. However, the variation in length is also large for Ni20, containing nanorods with both a length of over 700 nm and those with a length of only a few dozen nm. While in Ni0 in the image, only pure goethite nanorods can be seen, Ni10 and Ni20 show the presence of low crystalline as well as nanorods. The low crystalline material is thought to originate from ferrihydrite or amorphous by the Mössbauer spectra results. The low crystalline material also seems to change the amount of the specific surface area.

### 2.7. BET and BJH Analysis of α-Ni_x_Fe_1−x_OOH NPs

The experiment involved varying the relative pressure (*P*/*P*_0_), where *P* represents the adsorption equilibrium pressure, and *P*_0_ represents the saturation vapour pressure within the range of 0 to 1. The number of gas molecules adsorbed was quantified and graphed against the relative pressure, resulting in an isotherm. Figure 9 displays the isotherms of α-Ni_x_Fe_1−x_OOH NPs. Many factors influence the shape of the curve of an isotherm, such as the existence and dimensions of the pores and adsorption energy. In 1985, the International Union of Pure and Applied Chemistry (IUPAC) published a classification system comprising six types of adsorption/desorption isotherms. Over the past three decades, two additional types have been introduced. The isotherm shape observed for Ni0 and Ni10 is classified as type II according to this classification. Type II indicates the absence of pores or the potential presence of macropores (pores with sizes exceeding 50 nm) [36]. The IV and V types exhibit a unique phenomenon known as hysteresis, wherein the adsorption and desorption processes do not align as typically [37].

Hysteresis, closely associated with capillary condensation, primarily occurs in the mesopore range [38]. The hysteresis pattern observed can offer insights into the shape and structure of the pore. However, due to the diverse nature of hysteresis patterns, establishing a direct relationship between them and pore shape and structure is challenging.

On the other hand, the adsorption isotherms for Ni20, Ni30, Ni40, and Ni50 fall into the H4-type hysteresis classification. Type H4 is indicative of the presence of slit-type pores. This type of hysteresis pattern can sometimes be observed when micropores (pores with sizes of 2 nm or less) are present, as seen in type I isotherms [39]. Based on the isotherm shape depicted in Figure 9, it can be inferred that Ni0 to Ni10 possess macropores, while Ni20 to Ni50 exhibit micropores. The presence of micropores can also be deduced from the rapid increase in adsorption at low relative pressures and the characteristics of the hysteresis loop. The BET method can extract information regarding the adsorption process, specifically from monolayer adsorption to multilayer adsorption. The sample’s surface area can be determined by accurately calculating the degree of monolayer adsorption.

This can be achieved by multiplying the quantity of single-molecule adsorption by the cross-sectional area occupied by a single gas molecule, as follows [37].
(1)$$\;\frac{P}{V\left(P_{0}-P\right)}=\frac{1}{V_{m}C}+\left(\frac{C-1}{V_{m}C}\right)\left(\frac{P}{P_{0}}\right)$$

*P*, *P*_0_, *V_m_*, and *C* are parameters related to adsorption equilibrium pressure, saturation vapour pressure line, monolayer adsorption volume (e.g., adsorption volume when the gas molecules form a monolayer on a solid surface), and adsorption heat, respectively. The established relationship between *P*/*P*_0_ values in the range of 0.05 to 0.35 reveals significant insights.

The surface area, determined by applying the BET method, is presented in Table 5. Based on these calculations, the surface area of Ni0 was found to be 45.1 m^2^ g^−1^. With an increase in the Ni content from Ni0 to Ni20, the surface area progressively rises, reaching the highest value of 174.0 m^2^ g^−1^ for Ni20. However, as the Ni content is further increased, the surface area decreases, reaching 96.3 m^2^ g^−1^ for Ni40. Based on this result, it can be expected that the electrochemical properties of Ni20 will be the best due to possessing the largest surface area compared to all prepared samples.

Figure 9b shows that the average BJH pore diameter was ~1.9–3.9 nm, which demonstrates that the α-Ni_x_Fe_1−x_OOH NPs comprise micro- and mesopores as per the IUPAC definition. The BJH pore size distribution of all the α-Ni_x_Fe_1−x_OOH NPs shows the major existence of pores within 5 nm. The pore volume of α-Ni_x_Fe_1−x_OOH NPs samples was 0.60, 0.59, 0.51, 0.39, 0.44, 0.19, 0.11, 0.15 cm^3^ g^−1^ with ‘x’ of 0, 0.05, 0.10, 0.15, 0.20, 0.30, 0.40, and 0.50, respectively.

### 2.8. Bandgap Energy Derived from DRS of α-Ni_x_Fe_1−x_OOH NPs

For the bandgap (*E*_g_) estimation of powder samples, diffuse reflectance UV-Vis spectroscopy can be applied [40,41]. In this case, the detected light is reflected from the surface of the sample. The diffuse reflectance spectra (DRS) of α-Ni_x_Fe_1−x_OOH NPs are shown in Figure 10a. As for the DRS spectrum of Ni0, we can observe strong absorption at a wavelength of less than 380 nm, which relates to ligand-to-metal charge transitions [42]. A shoulder peak observed at 490 nm is attributed to two-step excitation, while weak absorption peaks of 610 and 890 nm are found due to the ligand field transition [42]. The relative absorption intensities of Ni5, Ni10, Ni15, Ni20, Ni30, Ni40, and Ni50 are larger than that of Ni0, indicating that Ni-doped FeOOH NPs absorbed the photon energy. *E*_g_ is an attribute of the photocatalyst, establishing its wavelength active region. It can be revealed using transmission spectra in thin films or absorption for bulk or powder samples dispersed in a solution.

The spectra obtained were analyzed using the Kubelka–Munk K-M function [40]. *E*_g_ represents the energy difference between the valence band’s top and the conduction band’s bottom level [43,44] and is a crucial parameter in assessing photocatalytic activity. *E*_g_ can be determined by constructing a Tauc plot [41,45]:(2)$${\left(h\nu \alpha \right)}^{1/n}=A\left(h\nu -E_{g}\right)$$
where *h* is Planck’s constant, *ν* is the frequency, *α* is the absorption coefficient, *A* is the proportionality constant, *E_g_* is the band gap energy, and *n* is a parameter depending on the type of transition in the material. For direct allowed transitions, it is taken as *n* = 1/2; for indirect allowed transitions *n* = 2 [46].

Figure 10 displays the plot of (*hνα*)^2^ versus (*hν*) obtained from diffuse reflectance spectroscopy (DRS) measurements of Ni_x_Fe_1−x_OOH NPs, enabling the estimation of the *E*_g_ values. The *E*_g_ value for pure goethite (Ni0) was determined to be 2.71 ± 0.01 eV. Table 5 shows that the *E*_g_ values for all Ni-doped samples were smaller than that of Ni0. Notably, the smallest *E*_g_ value of 2.06 ± 0.01 eV was observed for the Ni30 sample, determined as the *x*-axis intercept of the fitted straight lines depicted in Figure 10b. The *E*_g_ values of Ni40 and N50 were increased to 2.13 ± 0.01 eV and 2.15 ± 0.01 eV, respectively. The existence of Ni(OH)_2_ precipitated in Ni100x with x larger than 30 was the reason for the increase in band gap energy because NiO_6_ in Ni(OH)_2_ has a similar environment to NiO [36].

This comparison highlights the greater effectiveness of Ni addition in reducing the band gap. Photogenerated carriers are produced when a photocatalyst absorbs photons with energy larger than its *E*_g_. Consequently, in this study, the Ni_x_Fe_1−x_OOH NPs exhibit high excitability when exposed to a metal–halide lamp emitting light with wavelengths ranging from 250 to 750 nm. This excitation greatly enhances the photocatalytic performance [47]. In conclusion, the above observations validate that the Ni_x_Fe_1−x_OOH NPs photocatalysts investigated in this study possess an appropriate band gap and effectively utilize light.

### 2.9. Photo-Fenton Catalytic Ability of α-Ni_x_Fe_1−x_OOH NPs

The reaction rate constant *k* of methylene blue decomposition is calculated using the following Equation (3) [14]:(3)$$\mathrm{Ln}\left(c_{t}/c_{0}\right)=-kt$$

*c*_0_ is the concentration before the photocatalytic reaction of MB*_aq_* (=20 µM), and *c_t_* means the concentration of MB*_aq_* at time *t*.

Figure 11a shows the photo-Fenton reaction tested in the dark between Ni_x_Fe_1−x_OOH NPs samples and MB*_aq_*. It is noted that no decrease in the MB*_aq_* appeared for the degradation test using Ni0, and a slight degradation showed for Ni30. In contrast, increasing the Ni content to 0.50 shows a reduction in MB*_aq_*.

After the degradation test was conducted without light, a continuous photo-Fenton reaction test was carried out using MB*_aq_* and α-Ni_x_Fe_1−x_OOH NPs nanoparticles under visible light. The corresponding plots depicting the variation of (*c*/*c*_0_) over time (t) under visible light conditions are presented in Figure 11b. The most rapid decrease in (*c*/*c*_0_) during the photo-Fenton reaction is observed with Ni10. This can be attributed to the oxidation of Ni that occurred in this sample. This sample has the highest Ni-*K* XAFS absorption energy, which leads to it being the Ni10 containing the most Ni(III), see Figure 4c.

Figure 11c illustrates the relationship between ln(*c_t_*/*c_0_*) and t for α-Ni_x_Fe_1−x_OOH NPs. The values of *k*, representing the rate constant for MB*_aq_* decomposition, were determined and are presented in Table 5. For Ni0, the value of *k* is 6.4 ± 0.1 × 10^−3^ min^−1^, while the maximum value of *k*, 15.8 ± 0.6 × 10^−3^ min^−1^, is observed for Ni10. As Ni is added to goethite, the *k* values show an increasing trend from Ni5 to Ni10. However, for Ni15 and the subsequent samples, the *k* values are smaller than that for Ni0.

In the case of Ni15 and the later samples, a decrease in goethite content is observed, accompanied by the generation of a significant amount of low-crystalline substances. This change in composition is believed to be the underlying cause of the lower *k* values observed. Notably, Ni10 exhibits the highest *k* value among the samples. This can be attributed to the findings from the DRS, XAFS, and FeMS data, which suggest a reduction in the band gap and non-oxidation processes occurring in the Ni10 sample.

### 2.10. Electrical Properties—Solid-State Impedance Spectra and DC Conductivity

The experimental data obtained through solid-state impedance spectroscopy (SS-IS) are presented in a complex impedance plane, known as the Nyquist diagram, see Figure 12a–c and Appendix A. The analysis of these plots involves employing electrical equivalent circuit (EEC) modelling, which utilizes a complex nonlinear least-square (CNLLSQ) fitting procedure. It is apparent that the impedance spectra observed in all examined samples exhibit a distinct semicircle, representing the bulk electrical process within the investigated α- Ni_x_Fe_1−x_OOH NPs.

This behaviour can be effectively described by an equivalent R-CPE circuit. The parameters for each circuit element (*R*, *A*, and *α*) were directly derived from the measured impedance data using the CNLLSQ method.

In the present study, we took a step further and examined the conductivity spectra of all the samples. The conductivity spectra of samples with Ni10 and Ni20 at different temperatures are shown in Figure 13a,b, respectively. Although the conductivity isotherms have a similar shape, overall spectral features can be observed as follows. First, there exists a frequency-independent conductivity plateau at low frequencies. This particular feature is associated with the long-range transportation of charge carriers and represents the overall resistance observed in the impedance spectra or DC conductivity. In addition, frequency-dependent conductivity, commonly referred to as conductivity dispersion, manifests itself with increasing frequency in the form of a power-law.

This behaviour arises from localized movements of charge carriers occurring over short distances. We used values of the fitting parameter *R* obtained from modelling along with sample geometry to determine the total DC conductivity, as shown in Table 6. The DC conductivity obtained is in good correlation with the observed DC plateaus in the conductivity spectra; see Figure 13a,b.

The DC conductivity in our samples demonstrates a temperature-dependent behaviour that follows an Arrhenius relationship, indicating semiconducting characteristics, see Figure 13c.

Consequently, the activation energy for the DC conductivity, *E_DC_*, was determined for individual samples from the slope of log(*σ_DC_*) versus 1000/T using the equation:(4)$$\;\sigma_{DC}=\sigma_{0}^{*}\exp\left(\frac{-E_{DC}}{k_{B}T}\right)$$
where *σ_DC_* is the DC conductivity, $\sigma_{0}^{*}$ is the pre-exponent, *k_B_* is the Boltzmann constant, and *T* is the temperature (K). The activation energy, *E_DC_*, and DC conductivity, *σ_DC_*, at 90 °C for all investigated α-Ni_x_Fe_1−x_OOH NPs samples are presented in Table 6.

Continuing our analysis, we turn the focus to the comparison of the conductivity spectra at 110 °C, shown in Figure 14a. First, as previously mentioned, the shape of the conductivity spectra does not change with the composition. This consistency indicates that the mechanism of electrical transport remains unaffected. However, the modification of α-Ni_x_Fe_1−x_OOH NPs and the increase in Ni content has a negative effect, resulting in a decrease in DC conductivity, see Figure 14b and Table 6. As the Ni/(Fe + Ni) ratio increases, the DC conductivity exhibits a nearly linear decline, from 5.52 × 10^−10^ (Ω cm)^−1^ for Ni10 to 5.30 × 10^−12^ (Ω cm)^−1^ for the Ni50 sample. Conversely, the activation energy for DC conductivity, *E*_DC_, follows the opposite trend, with values increasing in the 73.0–82.5 kJmol^−1^ range.

However, it is important to note that our understanding of the electronic structure of iron oxides and oxyhydroxides, including goethite, remains incomplete. In our study on Ni_x_Fe_1−x_OOH NPs, the obtained values for the activation energy (0.75–0.85 eV), see Table 5, are almost three times lower in comparison to the goethites studied by Guskos et al. [48]. Our findings align more closely with the activation energy values reported for magnetite and hematite. Similar values are observed in the literature for various materials with disordered or partially disordered structures and dominant electron transport [49,50,51,52].

The goethite structure can be described as parallel double chains of edge-sharing octahedra. These chains consist of Fe^III^ bonded to three oxide ions and three hydroxides extending along the [001] direction and are linked to the neighbouring double chains by corner sharing. Guskos et al. [48] conducted an electrical study on goethite and proposed that charge transport occurs through thermally activated three-dimensional hopping of electrons via oxygen vacancies, based on DC electrical measurements conducted over a wide temperature range.

Furthermore, Vitaly et al. [53] showed that RT charge transport in goethite is primarily governed by the thermally activated hopping of small polarons, with the associated mobility being higher compared to other iron oxyhydroxide (FeOOH) polymorphs. Small polarons are formed when electrons self-trap onto an iron centre, resulting in conduction through phonon-mediated hops between centres [54]. Different inequivalent paths for electron hopping characterized by different Fe-Fe bond distances and species bridging two neighbouring Fe atoms are identified [53].

The pathway involving migration along the double chain ([001] direction) through shared octahedral edges, with electron transport mediated by O and OH species, is characterized by Fe atoms with parallel spins and with the shortest Fe^2+^-Fe^3+^ distance of approximately 3 Å. Su et al. [55] studied the electrical conduction mechanism of goethite under pressures up to 17.1 GPa using impedance spectroscopy. The results indicate a pressure-induced conduction mechanism transition around 5 GPa from mixed protonic-electronic conduction to pure electronic conduction, which is associated with the pressure-induced magnetic state transition.

In our study, we used the IS method in an inert atmosphere (N_2_), so the protonic contribution is expected to be inhibited and does not contribute to the total conductivity. Thus, the obtained trend in our study could lead to the conclusion that as the Ni content is increasing, the structure and bonding are affected in the studied Ni_x_Fe_1−x_OOH NPs samples, which leads to the increase in polaron hopping activation energies, thus, decreasing the ground state carrier mobility compared to pure goethite. Moreover, at the same time, with Ni doping and a decrease in Fe content, the charge carrier concentration decreases. Additionally, the presence of mixed metal centres (Fe, Ni) does not appear to have a favourable effect on polaron transport. Overall, the effects above collectively contribute to a decrease in the DC conductivity as the Ni content increases in the studied Ni_x_Fe_1−x_OOH nanoparticles.

### 2.11. The Electrochemical Properties of α-Ni_x_Fe_1−x_OOH NPs/Li- and Na-Ion Batteries

Table 7 shows the discharge capacity and capacity retention of Ni0, Ni10, Ni20, and Ni30 samples as a cathode in LIB under a low current rate at 5 mA g^−1^; the initial discharge capacity of Ni0 was the highest, reaching 1069 mAh g^−1^. The discharge capacity decreased with the increase in Ni content to 250 mAh g^−1^ for Ni30. Moreover, the capacity retention of all samples suddenly dropped after 10 cycles. The capacity retention of Ni0, Ni20, and Ni30 was less than 1%, while Ni10 was 7.4%.

On the other hand, after a tenfold increase in the current rate to 50 mA g^−1^, the initial discharge capacity of Ni0, Ni10, and Ni20 was 333, 350, and 494 mAh g^−1^, respectively, and the discharge capacity increased with the Ni content.

However, capacity retention decreased when the Ni content was increased. We can conclude that after increasing the current rate from 5 to 50 mA g^−1^, the initial capacity of all samples decreased, while the capacity retention was enhanced, as shown in Table 6 and Figure 15a.

In contrast, the discharge capacities of Ni0, Ni10, Ni20, Ni30, and Ni50 were 110, 116, 223, 189, and 202 mAh g^−1^, respectively, which was proportional to the size of the surface area as shown in Table 4 [56,57].

Additionally, the capacity retention after 30 cycles of all samples after adding Ni was lower than that for Ni0, as shown in Table 8 and Figure 15b. According to the aforementioned results concerning α-Ni_x_Fe_1−x_OOH NPs as cathode material in both LIB and SIB, with a specific focus on capacity retention after 30 cycles, one can deduce that the performance of Ni_x_Fe_1−x_OOH NPs as active cathode materials is more comparable in SIB compared to LIB.

## 3. Materials and Methods

### 3.1. Preparation of α-Ni_x_Fe_1−x_OOH NPs

For preparing the α-Ni_x_Fe_1−x_OOH NPs, iron (III) chloride hexahydrate (Wako, Japan, 095-00875) and nickel (II) chloride (Wako, Japan, 141-01062) were dissolved in 30 mL of pure water under magnetic stirring. The concentration and molar ratio of nickel are presented in Table 9. Once the mixture solution was thoroughly homogenized, 3 mol L^−1^ sodium hydroxide (Wako, Japan, 194-02135) aqueous solution was added until the pH reached 13.

The mixture solution underwent ultrasonication for 20 min after vigorous stirring. Furthermore, the mixture solution was stirred for an additional 20 min. The obtained mixture was then transferred into a Teflon-lined stainless-steel autoclave and maintained at 80 °C for a hydrothermal reaction period of 24 h. After naturally cooling to RT, the solid products were washed with pure water and ethanol to remove the neutral electrolyte.

The solid products were dried at 60 °C for 12 h and allowed to cool naturally. These samples were, respectively, abbreviated as Ni0, Ni5, Ni10, Ni15, Ni20, Ni30, Ni40, and Ni50.

### 3.2. Structural Characterization

The X-ray diffraction (XRD) patterns were measured by using a RINT-TTR III (Rigaku) X-ray diffractometer with Cu-Kα: λ = 1.54 Å. The measurements were conducted over a 2 θ range of 10° to 80°, with a data interval and scanning rate of 0.02° and 5° min^−1^, respectively. The X-ray generator was operated at 50 kV and 300 mA. The obtained XRD patterns were analyzed using Smartlab Studio II Powder XRD software, version 4.2.137.0 with the database of ICDD.

Thermogravimetry differential thermal analysis (TG-DTA) was performed by (Thermo plus TG8120, Rigaku, Japan), under a heating rate of 10 K min^−1^ and the temperature range (RT-1000 °C). The weight of the α-Ni_x_Fe_1−x_OOH NPs and α-Al_2_O_3_ reference was fixed to be 10 mg.

The Fourier transform infrared spectroscopy (FTIR) transmission spectra of α-Ni_x_Fe_1−x_OOH NPs were recorded by the (FTIR PerkinElmer, Shelton, CT, USA; spectral resolution: 1 cm^−1^) spectrometer in the range 400–4000 cm^−1^. The α-Ni_x_Fe_1−x_OOH NPs samples as a powder were mixed with KBr in a pellet of 10 mm diameter under a pressure of 90 kg/mm^2^ (KBr pellets technique), with the ratio of α-Ni_x_Fe_1−x_OOH NPs sample weight to KBr at ~1%.

The ^57^Fe Mössbauer spectra were measured at RT and at 86 K by a constant acceleration method with a ^57^Co(Rh) source, and α-Fe was used as a standard reference material. The obtained spectra were analyzed using the Mosswinn 4.0 software. The RT–X-ray absorption spectra of Fe*-K* and Ni*-K* (XANES/EXAFS) were measured in transmission mode by a beamline BL-12C at the High Energy Accelerator Research Organization (KEK-PF, 1-1 Oh-ho, Tsukuba, Ibaraki, Japan). The X-ray beam emitted from the synchrotron was monochromatized using a Si(111) double crystal and further attenuated to suppress the higher harmonics by employing a Ni mirror. The intensity of the X-ray beam was measured by setting an ionization box before and after transmission. In the front chamber, a mixture of N_2_+He gases (N_2_: 30%, He: 70%) was filled, while in the rear chamber, a mixture of Ar+N_2_ gases (Ar: 30%, N_2_: 70%) was utilized for the X-ray intensity measurements. The required amount of the sample needed for an excellent spectrum was calculated by SAMPLEM4M software version 0.9.71. The sample and boron nitride were mixed and homogenized in a mortar for 20 min, followed by pressing to make pellets with a diameter of 1.0 cm. The X-ray absorption spectra were analyzed by Athena software version 0.9.26. Transmission electron microscopy (TEM) images were obtained using a JEM-3200FS Field Emission Energy Filter Electron Microscope (JEOL, Tokyo, Japan). The optical bandgap was recorded with a Shimadzu UV-3600 spectrometer with an integrating sphere attachment (ISR-3100, Shimadzu, Kyoto, Japan). Diffuse reflectance UV-Vis spectroscopy, combined with the Kubelka–Munk equation and Tauc plots, were utilized to estimate the optical band gap. The specific surface area (SSA) was measured by BERSORP MINI X (MICRO TRAC BEL, Osaka, Japan) with N_2_ gas as the adsorbate, and the SSA was calculated by the Brunauer–Emmett–Teller (BET) method. The preprocessing was performed at 60 °C for 24 h using the BELSORP VAC II (MICRO TRAC BEL, Osaka, Japan). The obtained results were analyzed by BELMaster7(MICROTRAC BEL).

### 3.3. Photocatalytic Activity

To evaluate the photocatalytic performance of the α-Ni_x_Fe_1−x_OOH NPs, the degradation of methylene blue (MB) in an aqueous solution was measured, combining the produced sample (10 mg), MB (Wako, Japan, 7220-79-3) (20 µM, 10 mL), and H_2_O_2_ (Wako Japan, 081-04215) (9.75M, 40 µL). The UV-Vis spectra of the MB aqueous solution were measured by a GENESYS^TM^ 10S UV-Vis spectrophotometer in a wavelength region from 200 to 800 nm every 10 min with photoirradiation and stirring. The concentration of the MB aqueous solution after each interval was measured using the absorbance at the wavelength of 665 nm. The visible-light Fiber-Lite MH-100 metal–halide lamp emitted a wavelength range from 250 to 750 nm, and the output power was 200 W.

### 3.4. Solid-State Impedance Spectroscopy (SS-IS) of α-Ni_x_Fe_1−x_OOH NPs

The electrical properties were studied by solid-state impedance spectroscopy. Powder samples were pressed into cylindrical pellets 10 mm in diameter and with a thickness of approximately 0.5 mm under a uniform load of 2 tons using a hydraulic press. Gold electrodes 6 mm in diameter were sputtered onto both sides of the sample pellets using Sputter coater SC7620, Quorum Technologies for the electrical contact. The complex impedance was measured using an impedance analyzer (Novocontrol Alpha-AN Dielectric Spectrometer, Novocontrol Technologies GmbH & Co. KG, Montabaur, Germany) over a wide frequency range from 0.01 Hz to 1 MHz at temperatures between 30 °C and 170 °C (step 20 °C). The temperature was controlled to an accuracy of ±0.2 K.

The experimental data were analyzed by the electrical equivalent circuit (EEC) modelling employing the complex nonlinear least-square (CNLLSQ) fitting procedure. Depending on the EEC used and the obtained fitting parameters, various process(es) can be identified and separated. Typically, a single impedance semicircle can be represented by an EEC that combines a resistor and a capacitor connected in parallel. Ideally, this semicircle should pass through the origin of a complex plot and yield a low-frequency intercept on the real axis, corresponding to the resistance, R, of the observed process. However, in cases where the experimentally observed semicircle appears depressed, an alternative component known as the constant-phase element (CPE) is used instead of the standard capacitor in the equivalent circuits. The CPE is an empirical impedance function of the type: Z*_CPE_ = 1/*A*(iω)^α^ where *A* and *α* are the constants. For the samples in this study, the complex impedance spectrum is described by an equivalent R-CPE circuit. The parameters for each circuit element (*R*, *A*, and *α*) were determined directly from the measured impedance data using the CNLLSQ method. The DC conductivity values are calculated based on the modelling process, considering both the parameters obtained and the sample geometry.

### 3.5. Preparation of SIB Containing α-Ni_x_Fe_1−x_OOH NPs Cathode

To prepare the cathode, first, 250 mg of α-Ni_x_Fe_1−x_OOH NPs and 95 mg of acetylene black (AB, Strem chemicals 06-0025, Newburyport, MA, USA) were mixed by a ball mill (Planet M2-3F, Nagao System, Kanagawa, Japan) at 800 rpm for 15 min. Then, 5 mg of polytetrafluoroethylene (PTFE, Wako, Japan, 165-13412) was added to the 95 mg ball-milled mixture powder and mixed in the mortar until the powder became semi-solid and then pressed in a pellet with a diameter of 1 cm and weight of 30 mg. For the sodium-ion battery, metallic Na (Kishida, Osaka, Japan, 750-70852) (90 mg) and an electrolyte of 1 M NaClO_4_/propylene carbonate solution (Tomiyama LIPASTE-P/S1, Tokyo, Japan) were used, while metallic Li (Wako, Japan, 127-06001) (30 mg) was used with LiPF_6_/Ethylene carbonate and Dimethyl carbonate 1:1 *v*/*v*% ratio (Kishida LBG-00022, Osaka, Japan) for the lithium-ion battery. The 2032-type coin battery was assembled in a glove box under an oxygen concentration less than 0.99 ppm. TOSCAT-3100SK (TOYO-SYSTEM, Fukushima, Japan) measured the charge–discharge capacity performance at 30 cycles under the voltage range of 0.8–4.0 V, a current rate of 5 and 50 mA g^−1^, and a lower limit current of 0.1 mA. This process started with the discharge process in discharge-CC and charge-CC/CV.5.

## 4. Conclusions

In this paper, we studied the relationship between local structure, photo-Fenton catalytic ability, and electrochemical properties in LIBs and SIBs of Ni-doped goethite with the composition of α-Ni_x_Fe_1−x_OOH NPs (x = 0, 0.05, 0.10,0.15, 0.20, 0.30, 0.40, and 0.50, abbreviated as Ni100*x). The XRD patterns of α-Ni_x_Fe_1−x_OOH showed the α-FeOOH crystalline phases in all prepared samples. Moreover, a new crystalline phase is related to Ni(OH)_2_, for Ni contents from 0.30 to 0.50. The weight loss at low-temperature increased in the Ni_x_Fe_1−x_OOH NPs with the increased Ni content, which can be explained by more adsorbed water in the NiFeOOH NPs samples with a large surface area. The transmission peak observed at 3130 cm^−1^ corresponds to the O-H stretching vibration of the hydroxyl group. The strong two transmission peaks were observed at 797 cm^−1^ and 895 cm^−1^, related to Fe-O-H bending vibrations out of the plane and in the plane, respectively. The absorption edge energy of α-Ni_x_Fe_1−x_OOH NPs is higher than the values of Fe_2_O_3_ (Fe^3+^); this implies that none of the α-Ni_x_Fe_1−x_OOH NPs samples contained Fe^2+^, and they contained only Fe^3+^. Moreover, the *Ni-K* absorption edge energy of Ni_x_Fe_1−x_OOH NPs is higher than that expected for pure Ni^2+^. Samples Ni15 and Ni20 contain less Ni^3+^, while Ni10 has the highest Ni^3+^ content. The ^57^Fe Mössbauer spectrum of Ni0, measured at RT, displayed a sextet corresponding to goethite with an isomer shift (*δ*) of 0.36 mm s^−1^ and a hyperfine magnetic distribution (*B*_hf_) of 32.4 T. The absorption area of the superparamagnetic component increased from 4.8% to 82.9%, with an increase in Ni content up to an ‘x’ value of 0.2. The surface area of Ni100*x changed from 45.1 to 59.9, 73.9, 135, 174, 145, and 178 m^2^ g^−1^ with ‘x’ 0, 0.05, 0.10, 0.15, 0.20, 0.30, 0.40, and 0.50, respectively. In addition, the optical band of Ni_x_Fe_1−x_OOH NPs decreased from 2.71 eV to 2.15 eV with the increasing Ni content from 0 to 0.50. The largest first-order rate constant (*k*) of 14.6 × 10^−3^ min^−1^ was measured for Ni10. The DC conductivity decreased from 5.52 × 10^−10^ to 5.30 × 10^−12^ (Ω cm)^−1^ with ‘x’ increasing from 0.10 to 0.50. The highest initial capacity was recorded at 494 mAh g^−1^ for Ni30 at the current rate of 5 mA g^−1^ as a cathode material in LIB. Meanwhile, in SIB, the largest initial capacity was found as 233 mAh g^−1^ for Ni20 at a current rate of 5 mA g^−1^. In conclusion, α-Ni_x_Fe_1−x_OOH NPs can be effectively utilized as visible-light-activated catalysts and active cathode materials in secondary batteries.

## Figures and Tables

**Figure 1 ijms-24-14300-f001:**
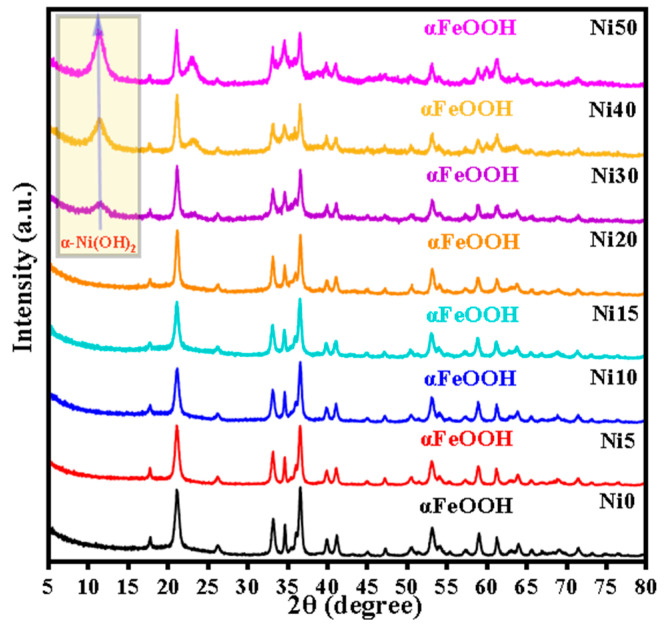
XRD pattern of α-Ni_x_Fe_1−x_OOH nanoparticles (NPs) with ‘x’ from 0 to 0.50.

**Figure 2 ijms-24-14300-f002:**
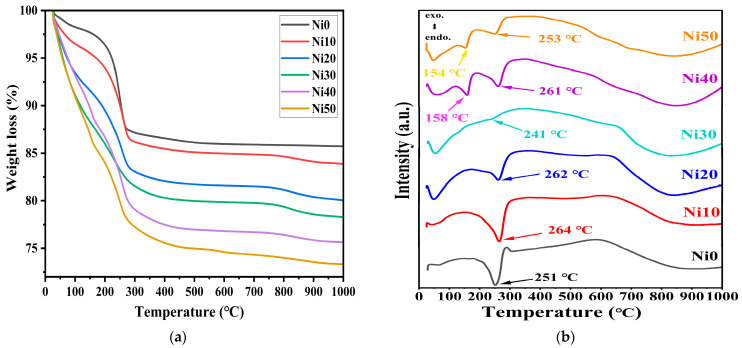
(**a**) Thermogravimetric (TG) and (**b**) differential thermal analysis (DTA) of α-Ni_x_Fe_1−x_OOH NPs with ‘x’ from 0.0 to 0.50.

**Figure 3 ijms-24-14300-f003:**
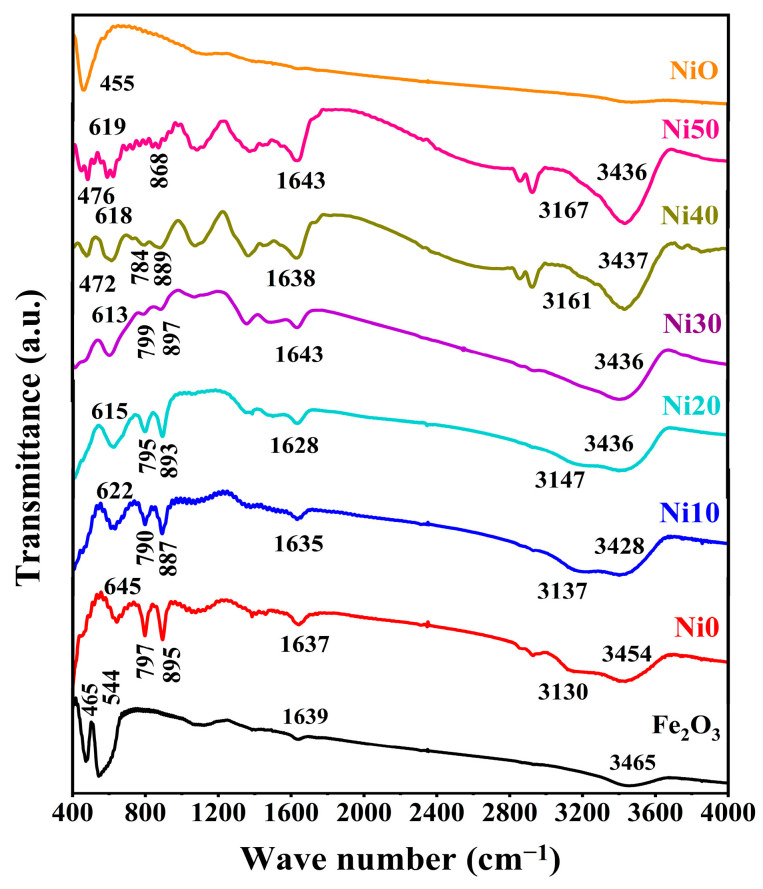
FTIR spectra of α-Ni_x_Fe_1−x_OOH NPs with ‘x’ from 0.0 to 0.50.

**Figure 4 ijms-24-14300-f004:**
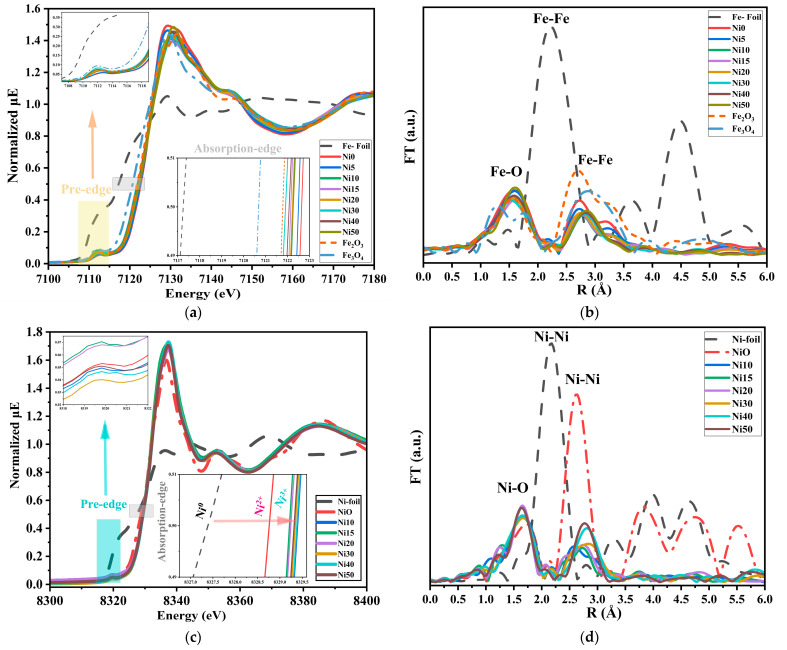
Fe*-K* and Ni*-K* edge XANES and FT-EXAFS spectra of α-Ni_x_Fe_1−x_OOH NPs with ‘x’ from 0.0 to 0.50 together with references of Fe-foil, α-Fe_2_O_3_, Ni-foil, and NiO, (**a**) Fe*-K* edge XANES, (**b**) Fe*-K* FT-EXAFS, (**c**) Fe*-K* FT-EXAFS, and (**d**) Ni*-K* FT-EXAFS.

**Figure 5 ijms-24-14300-f005:**
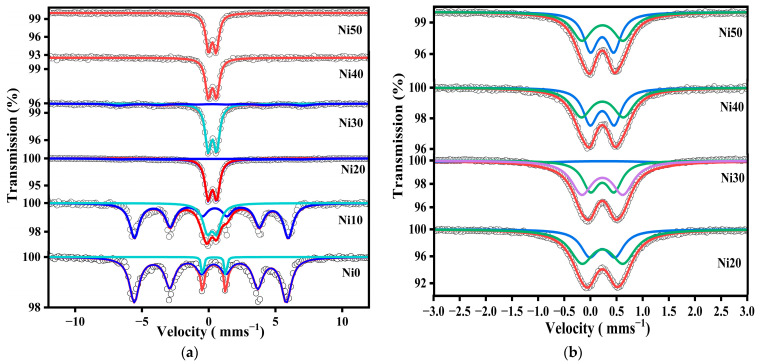
^57^Fe-Mössbauer spectra of α-Ni_x_Fe_1−x_OOH NPs with ‘x’ from 0.00 to 0.50 measured at room temperature in the velocity range of (**a**) ±12 mm s^−1^ and (**b**) ±3 mm s^−1^.

**Figure 6 ijms-24-14300-f006:**
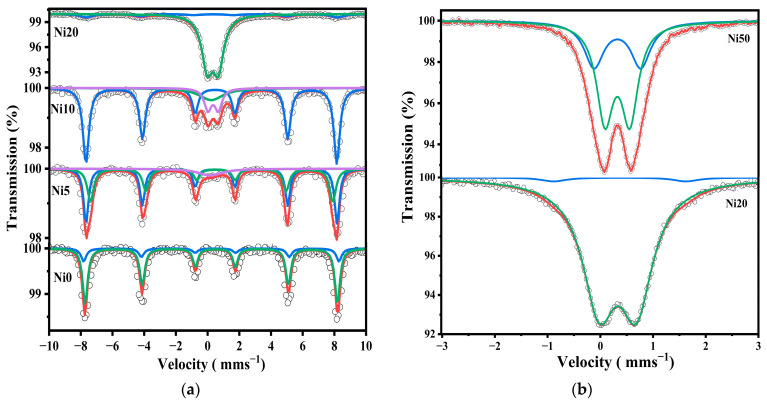
^57^Fe-Mössbauer spectra of α-Ni_x_Fe_1−x_OOH NPs nanoparticles measured at 86 K in the velocity range of (**a**) ±12 mm s^−1^, with ‘x’ from 0.00 to 0.20 and (**b**) ±3 mm s^−1^ with ‘x’ of 0.20 and 0.50.

**Figure 7 ijms-24-14300-f007:**
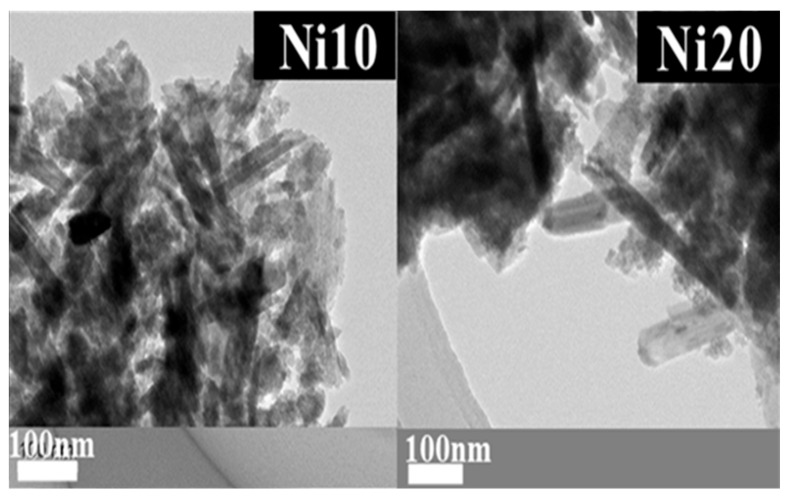
TEM images of α-Ni_x_Fe_1−x_OOH NPs with ‘x’ of 0.10 and 0.20.

**Figure 8 ijms-24-14300-f008:**
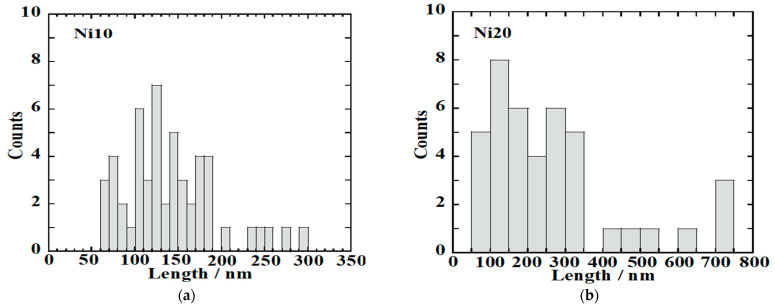
Histograms of the length of α-Ni_x_Fe_1−x_OOH NPs with ‘x’ of (**a**) 0.10 and (**b**) 0.20.

**Figure 9 ijms-24-14300-f009:**
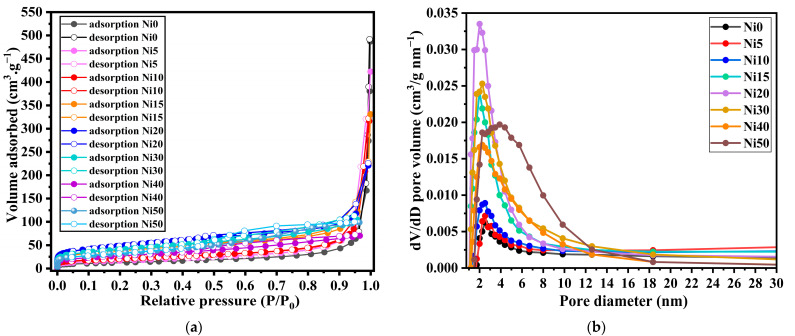
(**a**) N_2_ Adsorption/desorption isotherm and (**b**) the corresponding BJH pore size distribution curves of α-Ni_x_Fe_1−x_OOH NPs with ‘x’ of 0, 0.05, 0.10, 0.15, 0.20, 0.30, 0.40, and 0.50 using N_2_ as the adsorption material.

**Figure 10 ijms-24-14300-f010:**
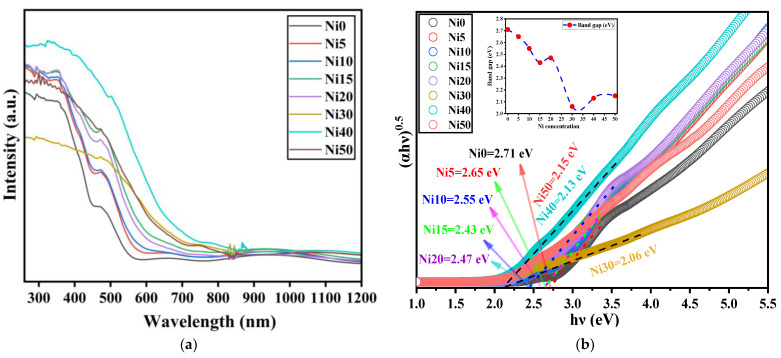
(**a**) UV-Vis diffuse reflectance spectra and (**b**) Tauc plot of α-Ni_x_Fe_1−x_OOH NPs with ‘x’ from 0 to 0.50. The dashed lines and arrows are drawn as guides for the eyes.

**Figure 11 ijms-24-14300-f011:**
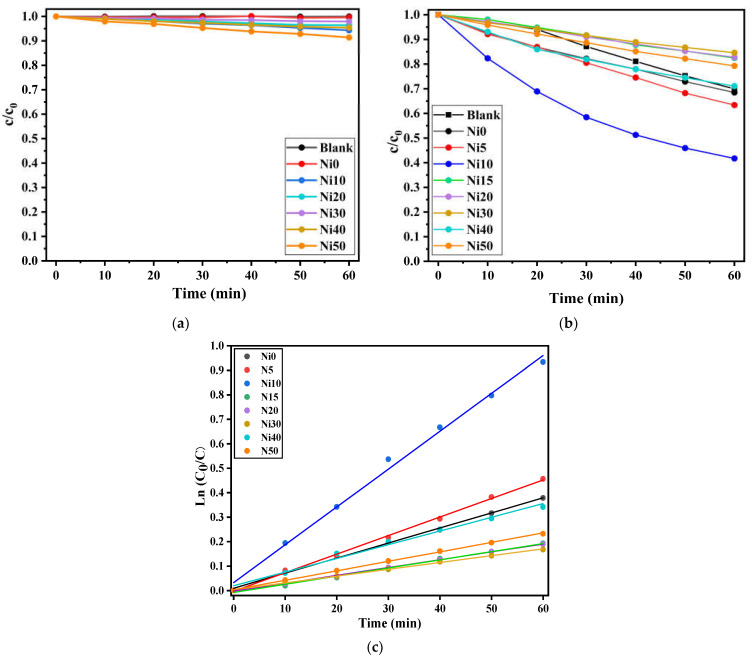
*c*/*c*_0_ versus *t* plots (**a**) in the dark and (**b**) under visible-light irradiation, and (**c**) ln (*c*_0_/*c*) versus *t* under the photo-Fenton reaction occurred by 20 µM MB*_aq_* in the aqueous solution and α-Ni_x_Fe_1−x_OOH NPs.

**Figure 12 ijms-24-14300-f012:**
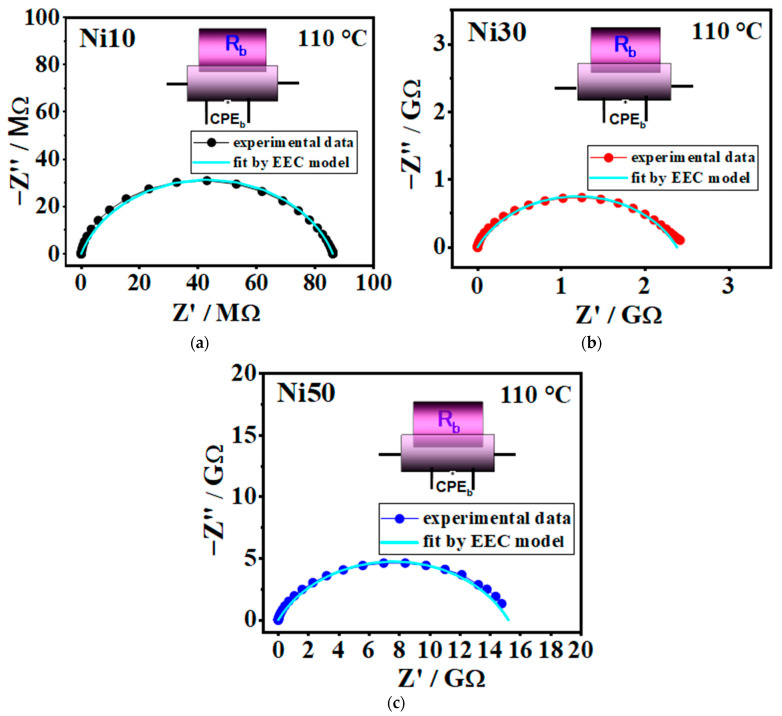
Complex impedance spectra @110 °C for samples (**a**) Ni10, (**b**) Ni30, and (**c**) Ni50. The symbols (coloured circle) denote experimental values, whereas the solid cyan line corresponds to the best fit. The corresponding equivalent circuit model, comprised of a parallel combination of the resistor (R) and the constant-phase element (CPE), is used for fitting the data of an individual spectrum, and its interpretation is shown in each figure (defined as follows: b-bulk phase).

**Figure 13 ijms-24-14300-f013:**
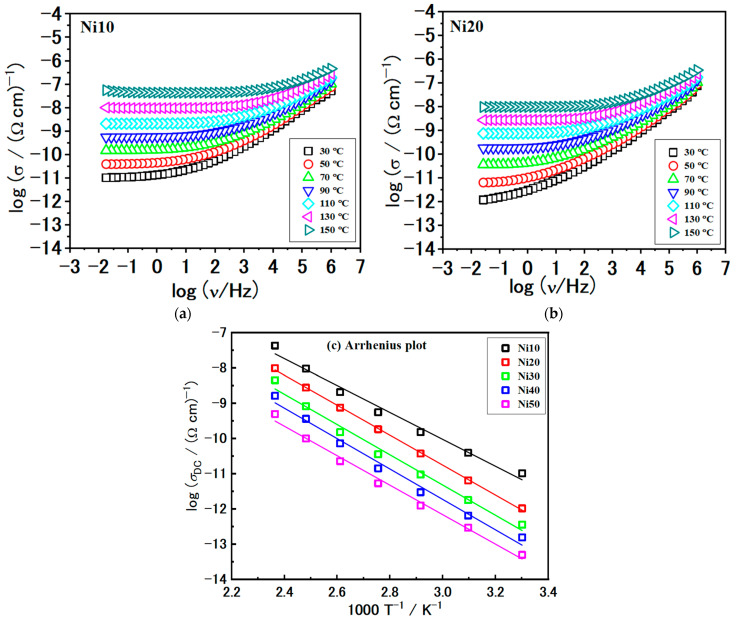
Conductivity spectra for sample (**a**) Ni10, (**b**) Ni20, and (**c**) Arrhenius plots of direct current (DC) conductivity (log(*σ_DC_*) versus 1000/*T*) for all studied samples. Solid lines in (**c**) represent the least-square linear fits to the experimental data. The error bars are, at most, of the order of the symbol size.

**Figure 14 ijms-24-14300-f014:**
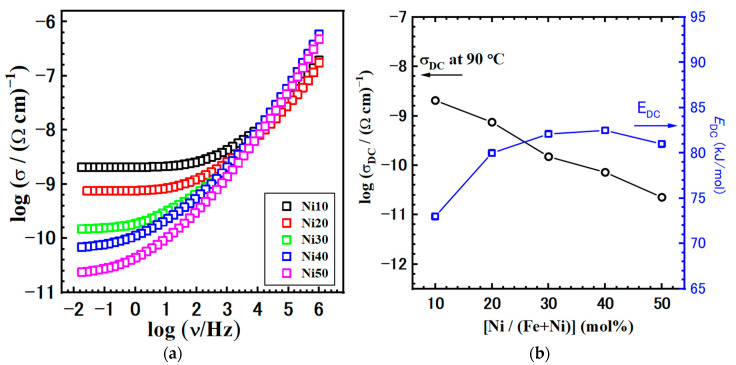
(**a**) Conductivity isotherms and (**b**) DC conductivity obtained at @110 °C with activation energy for DC conductivity, *E*_DC_, for all studied samples.

**Figure 15 ijms-24-14300-f015:**
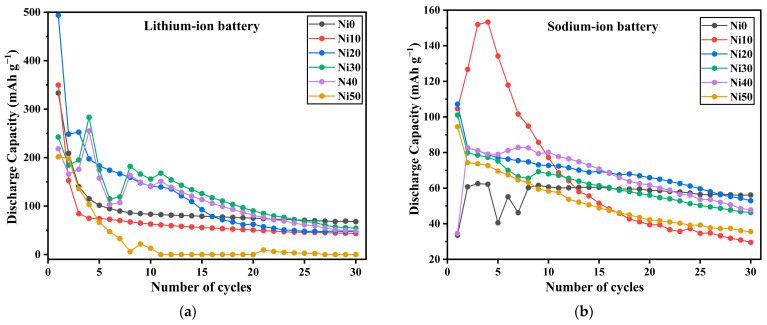
Charge–discharge curve and capacity recycling of cathode active performance for α-Ni_x_Fe_1−x_OOH NPs in (**a**) Li-and (**b**) Na-ion batteries evaluated under the current density of 0.15 mA and the voltage between 0.8 and 4.0 V.

**Table 1 ijms-24-14300-t001:** The major crystalline phase, lattice parameters, space group, and crystal structure of α-Ni_x_Fe_1−x_OOH nanoparticles.

Sample	Major Crystalline Phase	Lattice Parameters (Å)						*V* (Å^3^)	FWHM (110) (Deg.)	*d*_100_ (Å)	Space Group	% of Constituent Phases
		(Å)			(Degree)							
		a	b	c	α	β	γ					
Ni0	FeOOH	4.6215	9.9585	3.0249	90	90	90	143 (17)	0.587 (7)	4.1945 (4)	Pbnm	100
Ni5	FeOOH	4.6223 (1)	9.9565 (1)	3.0260 (1)	90	90	90	148 (19)	0.570 (7)	4.1957 (4)	Pbnm	100
Ni10	FeOOH	4.6255 (1)	9.9834 (2)	3.0273 (1)	90	90	90	148 (2)	0.572 (8)	4.1985 (5)	Pbnm	100
Ni15	FeOOH	4.6255 (1)	9.9999 (3)	3.0289 (1)	90	90	90	159 (4)	0.530 (13)	4.206 (2)	Pnma	100
Ni20	FeOOH	4.6240 (1)	9.9815 (2)	3.0287 (1)	90	90	90	194 (3)	0.436 (7)	4.1945 (5)	Pnma	100
Ni30	FeOOH	4.6330 (1)	9.9845 (2)	3.0245 (1)	90	90	90	168 (4)	0.503 (12)	4.184 (2)	Pnma	92.37
	αNi(OH)_2_	3.2006 (1)	3.2006 (1)	23.456 (5)	90	90	120	-	-	-	R3m:H	7.63
Ni40	FeOOH	4.6247 (1)	9.9792 (2)	3.0268 (1)	90	90	90	183 (4)	0.461 (11)	4.199 (19)	Pnma	91.93
	αNi(OH)_2_	3.1932 (1)	3.1932 (1)	23.445 (4)	90	90	120	-	-	-	R3m:H	8.07
Ni50	FeOOH	4.6240 (1)	9.9723 (2)	3.0242 (1)	90	90	90	173 (4)	0.487 (12)	4.199 (18)	Pnma	90.65
	αNi(OH)_2_	3.0972 (1)	3.0972 (1)	23.972 (7)	90	90	120	-	-	-	R3m:H	9.35

**Table 2 ijms-24-14300-t002:** Mössbauer parameters of α-Ni_x_Fe_1−x_OOH nanoparticles obtained from the ^57^Fe-Mössbauer spectra measured in the velocity range from 12 mm s^−1^ to −12 mm s^−1^ at room temperature.

Sample	Component	*A* (%)	*δ* (mms^−1^)	Δ (mms^−1^)	*Γ* (mms^−1^)	*B*_hf_ (T)	Phase
Ni0	sextet	95.2	0.36 ± 0.01	−0.26 ± 0.01	0.88 ± 0.01	32.95	Goethite
	doublet	4.8	0.53 ± 0.01	1.73 ± 0.01	0.19 ± 0.01	-	superparamagnetic
Ni10	sextet	74.4	0.43 ± 0.01	−0.26 ± 0.01	0.86 ± 0.01	28.23	Goethite
	doublet	25.6	0.34 ± 0.01	0.71 ± 0.01	0.78 ± 0.02	-	superparamagnetic
Ni20	sextet	17.1	0.33 ± 0.11	−0.13 ± 0.16	2.63 ± 0.28	27.84	Goethite
	doublet	82.9	0.38 ± 0.01	0.65 ± 0.01	0.55 ± 0.01	-	superparamagnetic
Ni30	sextet	19.2	0.31 ± 0.06	−0.10 ± 0.12	1.48 ± 0.17	27.16	Goethite
	doublet	80.8	0.38 ± 0.01	0.64 ± 0.01	0.53 ± 0.01	-	superparamagnetic
Ni40	doublet	100	0.38 ± 0.01	0.62 ± 0.01	0.52 ± 0.01	-	superparamagnetic
Ni50	doublet	100	0.38 ± 0.01	0.58 ± 0.01	0.49 ± 0.01	-	superparamagnetic

*δ*: isomer shift, Δ: quadrupole splitting, *Γ*: FWHM, *B*_hf_: hyperfine magnetic distribution.

**Table 3 ijms-24-14300-t003:** Mössbauer parameters of α-Ni_x_Fe_1−x_OOH NPs obtained from the ^57^Fe-Mössbauer spectra measured in the velocity range from 3 mm s^−1^ to −3 mm s^−1^ at room temperature.

Sample	Component	*A* (%)	*δ* (mms^−1^)	Δ (mms^−1^*)*	*Γ* (mms^−1^)	*B*_hf_ (T)	Phase
Ni20	sextet	2.61	0.25 ± 0.01	−0.10 ± 0.01	0.33 ± 0.26	27.72	Goethite
	doublet	50.25	0.34 ± 0.01	0.84 ± 0.04	0.53 ± 0.01	-	amorphous
	doublet	47.14	0.33 ± 0.01	0.51 ± 0.02	0.42 ± 0.02	-	Superparamagnetic
Ni30	sextet	15.72	0.29 ± 0.01	−0.14 ± 0.01	1.11 ± 0.21	27.08	Goethite
	doublet	51.81	0.33 ± 0.01	0.79 ± 0.02	0.48 ± 0.01	-	amorphous
	doublet	32.46	0.33 ± 0.01	0.45 ± 0.01	0.37 ± 0.01	-	Superparamagnetic
Ni40	doublet	42.27	0.33 ± 0.01	0.85 ± 0.03	0.46 ± 0.01	-	amorphous
	doublet	57.73	0.33 ± 0.01	0.48 ± 0.01	0.37 ± 0.01	-	Superparamagnetic
Ni50	doublet	39.45	0.33 ± 0.01	0.84 ± 0.02	0.44 ± 0.01	-	amorphous
	doublet	60.55	0.33 ± 0.01	0.47 ± 0.01	0.35 ± 0.01	-	Superparamagnetic

**Table 4 ijms-24-14300-t004:** Average length, width, and ratio (length/width) for α-Ni_x_Fe_1−x_OOH NPs with ‘x’ of 0, 0.10, and 0.20 evaluated from the TEM images.

Sample	Ave. Length (nm)	Ave. Width (nm)	Ave. Ratio (Length/Width)
Ni0	181 ± 83 [14]	21.6 ± 6.6 [14]	9.6 ± 6.5 [14]
Ni10	141 ± 54	33.2 ± 11.2	4.6 ± 2.0
Ni20	263 ± 179	42.9 ± 13.9	6.3 ± 3.6

**Table 5 ijms-24-14300-t005:** Specific surface area by BET method, pore diameter, pore volume, bandgap value, and *k* value of α-Ni_x_Fe_1−x_OOH NPs obtained from DRS and MB degradation test.

Sample	BET SSA (m^2^ g^−1^)	Pore Diameter (nm)	Pore Volume (cm^3^ g^−1^)	Bandgap (eV)	*k*/10^−3^ min^−1^
Ni0	45.1	2.51	0.60	2.71	6.2 ± 0.1
Ni5	59.9	2.51	0.59	2.65	7.6 ± 0.1
Ni10	73.9	2.51	0.51	2.55	14.6 ± 0.6
Ni15	135	1.99	0.39	2.43	3.3 ± 0.1
Ni20	174.0	1.99	0.44	2.47	3.2 ± 0.1
Ni30	145.0	2.24	0.19	2.06	2.8 ± 0.1
Ni40	96.3	2.24	0.11	2.13	5.6 ± 0.1
Ni50	117.3	3.88	0.15	2.15	3.95 ± 0.2

**Table 6 ijms-24-14300-t006:** DC conductivity, *σ_DC_;* activation energy, *E_DC_*; and pre-exponential factor, $\sigma_{0}^{*}$, for all studied glasses.

Glass	*σ_DC_* ^a^/ (Ω cm)^−1^	*E_DC_*/kJmol^−1^	$\sigma_{0}^{*}$/(Ω cm)^−1^
Ni10	5.52 × 10^−10^	73.0	1.42
Ni20	1.82 × 10^−10^	80.4	1.99
Ni30	3.56 × 10^−11^	82.1	1.55
Ni40	1.41 × 10^−11^	82.5	1.22
Ni50	5.30 × 10^−12^	80.5	0.39

^a^ values at 90 °C.

**Table 7 ijms-24-14300-t007:** Discharge capacity and capacity retention of thecharge–discharge capacity performance for α-Ni_x_Fe_1−x_OOH nanoparticles measured at the current rates of 5 and 50 mAh g^−1^ in LIB.

Sample	5 mAh g^−1^		50 mA mAh g^−1^	
	Discharge Capacity (mAh g^−1^)	Capacity Retention (%)	Discharge Capacity (mAh g^−1^)	Capacity Retention (%)
Ni0	1069	0.4	333	20.3
Ni10	363	7.4	350	12.4
Ni20	271	0.4	494	9.6
Ni30	250	0.9	-	-

**Table 8 ijms-24-14300-t008:** Discharge capacity and capacity retention of the charge–discharge capacity performance for a-Ni_x_Fe_1−x_OOH nanoparticles measured at the current rates of 5 and 50 mAh g^−1^ in SIB.

Sample	5 mAh g^−1^		50 mA mAh g^−1^	
	Discharge Capacity (mAh g^−1^)	Capacity Retention (%)	Discharge Capacity (mAh g^−1^)	Capacity Retention (%)
Ni0	110	31.7	-	-
Ni10	116	21.6	-	-
Ni20	223	12.7	107	49.4
Ni30	189	21.3	101	45.8
Ni50	202	27.7	95	37.6

**Table 9 ijms-24-14300-t009:** Fe and Ni concentrations of α-Ni_x_Fe_1−x_OOH nanoparticles synthesizing.

Sample Code	[Fe] (mol L^−1^)	[Ni] (mol L^−1^)	$\frac{\left[Ni\right]}{\left[Fe\right]+\left[Ni\right]}\mathrm{mol}\%$
Ni0	0.100	0	0
Ni5	0.095	0.005	5
Ni10	0.090	0.010	10
Ni15	0.085	0.015	15
Ni20	0.080	0.020	20
Ni30	0.070	0.030	30
Ni40	0.060	0.040	40
Ni50	0.050	0.050	50

## Data Availability

The data presented in this study are available from the corresponding author upon request.

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
