# Peer review of "Photocatalytic and Cathode Active Abilities of Ni-Substituted α-FeOOH Nanoparticles"

_ijms, 2023, doi:10.3390/ijms241814300_

Round 1

Reviewer 1 Report

The work is interesting and the authors have investigated both the photocatalytic activity and the electrochemical energy storage capability of Ni-doped FeOOH. However, there is a major need to revise the manuscript addressing the following comments before it can be re-evaluated for publication in IJMS.

Comment 1. The endothermic peak in the DTA curve for Ni 30 appears to have disappeared or highly diminished compared to the rest of the samples.  The reason should be mentioned in the manuscript.

Comment 2. There is no information for the SSA of Ni5, and Ni15 samples. Without these data, how can the authors do an analysis of the trend in the SSA of the samples with increasing Ni content?

Comment 3. The authors should include BJH curves for analyzing the pore sizes of the samples.

Comment 4. The Nyquist plots for only a few selected samples are provided. What about the rest of the samples? The authors should include all the samples for impedance spectra analysis.

Comment 5. The labeling in the Y-axis for Figures 13a-b should be corrected.

Comment 6. Both photocatalytic and electrochemical activities are greatly affected by morphological, chemical, and internal structural features. So, FE-SEM, XPS, and TEM characterizations should also be included to better understand the physico-chemical properties of the materials.

Comment 7. The changes in the phase, near chemical oxidation states, and the internal structure of the materials after their photocatalytic and electrochemical tests should also be investigated to understand the true nature of the active material.

Author Response

First of all, we would like to thank you for the nice and constructive comments on our paper.

Reviewer #1:

Q1:  The endothermic peak in the DTA curve for Ni 30 appears to have disappeared or highly diminished compared to the rest of the samples. The reason should be mentioned in the manuscript.

Response:

The authors thank the reviewer for his comment. In the revised manuscript, we modified and mentioned the reason for the high diminished compared to the rest of the samples in the endothermic peak for Ni30 in the DTA section.

Q2: There is no information for the SSA of Ni5, and Ni15 samples. Without these data, how can the authors do an analysis of the trend in the SSA of the samples with increasing Ni content?

Response:

The authors thank the reviewer for his comment. The SSA of Ni5 and Ni15 is included in the revised manuscript in Figure 9 (a) and Table 5.

Q3: The authors should include BJH curves for analyzing the pore sizes of the samples.

Response:

The authors thank the reviewer for his comment. The BJH curves were included in the revised manuscript, presented in Figure. 9 (b) and Table 5. The pore sizes of the samples were discussed in the revised manuscript.

Q4: The Nyquist plots for only a few selected samples are provided. What about the rest of the samples? The authors should include all the samples for impedance spectra analysis.

Response:

The authors thank the reviewer for his comment. We measured for all samples, and due to no change in the shape, we presented Ni10, Ni30, and Ni50 but not all. In supporting information, we included the Nyquist plots of Ni20 and Ni40 figures (Figure S2 (a, b). Also these two figures are presented below. 

Q5: The labeling in the Y-axis for Figures 13a-b should be corrected.

Response:

We thank the reviewer for his comment. The labeling in the Y-axis for Figures 13 a-b was modified in the revised manuscript.

Q6: Both photocatalytic and electrochemical activities are greatly affected by morphological, chemical, and internal structural features. So, FE-SEM, XPS, and TEM characterizations should also be included to better understand the physico-chemical properties of the materials.

Response:

We would like to thank the reviewer for their valuable comment. We added TEM analyisis to this manuscript. We are also working on detail micro(structural) effect on photocatalytic and electrochemical performance to understand physico-chemical properties of the such materials, and this is subject of our new separate ongoing work which we plan to finalized soon and submit. We belive it will shed more light on the correlations mentioned above.

Q7: The changes in the phase, near chemical oxidation states, and the internal structure of the materials after their photocatalytic and electrochemical tests should also be investigated to understand the true nature of the active material.

Response:

We would like to thank the reviewer for their valuable comment. As mentioned above, in the previous comment, detail micro(structural) effect on photocatalytic and electrochemical performance (before and after tests) to understand physico-chemical properties of the such materials, is subject of our new separate ongoing work which we plan to finalized soon and submit: We belive it will shed more light on the correlations mentioned above.

Reviewer 2 Report

The paper “Photocatalytic and cathode active abilities of Ni-substituted α-FeOOH nanoparticles” written by Ahmed Ibrahim and others presents study of local structure, photocatalytic ability, and cathode performances in sodium-ion batteries (SIBs) and lithium-ion batteries (LIBs) using Ni-substituted goethite nanoparticles (NixFe1-xOOH NPs) with a range of 'x' values from 0 to 0.5 ..

As can be seen from the title and abstract of the article, the authors proposed two different applications for their synthesized compounds as materials - photocatalysts and cathode materials. This is surprising since the requirements for these materials are different both in terms of physical properties and in terms of crystal structure. Nevertheless, the authors presented a large amount of experimental work both on the characterization of the obtained materials and on the study of their functional properties. Moreover, as follows from the abstract and the introductory part, they decided to reveal the relationship between nickel doping and the properties of the synthesized materials. It should be noted that the authors used the most modern methods of local structure analysis, including X-ray absorption spectroscopy (XANES/EXAFS) and 57Fe Mössbauer spectroscopy both at room and low temperatures.

At the same time, there are a number of comments to the text of the article.

1) In the introductory part, the properties and structure of the initial compound for doping, goethite, are described in sufficient detail and well. However, what exactly the authors expect from the replacement of iron by nickel in the goethite structure remains unclear. Is nickel valency important to them and why. And for each of the applications it is worth discussing separately.

2) Speaking about the results of X-ray diffraction, the authors note the absence of changes in the cell parameters for all concentrations, referring to the proximity of the radii of iron 3+ and nickel 2+. At the same time, according to X-ray absorption spectroscopy data, nickel is in an oxidation state greater than 2+. "According to the absorption edge's higher value compared to NiO, the oxidation state of Ni in all NixFe1-xOOH NPs samples is higher than +2." By the way, the question immediately arises: in what degree of oxidation more than 2+ can nickel be? Perhaps, to answer this question, it was necessary to measure standards not only with valence 0 and 2+, but also with valence 3+.

3) As for the hydrothermal synthesis of the compounds under study, it remains unclear from its description what causes the oxidation of nickel 2+ to nickel 3+ in a sealed hydrothermal cell during the synthesis of goethite. What was the oxidizing agent in this process?

4) When determining the band gap, the authors used the extrapolation of the linear part in energy coordinates. There are no explanations in the text how they chose this linear part, and this choice looks very arbitrary. This creates doubts about the reliability of the obtained data on the dependence of the band gap on the nickel concentration.

5) The photocatalytic properties of the synthesized compounds did not show a systematic behavior in a series of nickel concentrations. All samples show either the same properties as the original goethite, or even worse. The nickel10 sample alone showed an improvement, but there is no explanation in the text why this happened. By the way, it is not clear why the authors used hydrogen peroxide in the measurements. If it is a sacrificial reagent, then it would be necessary to give blank measurements without a catalyst in the presence of peroxide and in the presence of a catalyst without light.

6) In electrochemical experiments, no systematicity is observed either, and a sharp drop in the first cycles indicates the rapid degradation of all samples, which may be due to the presence of a large amount of adsorbed water and chemically bonded water. This is confirmed by TGA data. In this regard, the question arises as to the expediency of using such compounds as cathode materials. Again, the measurements performed by the authors of two-phase samples containing, according to XRD, nickel hydroxide, are surprising.

8) As for the conclusions, on the one hand they contain a lot of unnecessary information about not the most important results, on the other hand, there is no main conclusion about the role of the doped nickel for the studied properties of the synthesized compounds.

All these questions indicate the incompleteness of the work on the article and it cannot be accepted for publication in the submitted form.

No comments

Author Response

First of all, we would like to thank you for the nice and constructive comments on our paper.

Reviewer #2:

Q1: In the introductory part, the properties and structure of the initial compound for doping, goethite, are described in sufficient detail and well. However, what exactly the authors expect from the replacement of iron by nickel in the goethite structure remains unclear. Is nickel valency important to them and why. And for each of the applications it is worth discussing separately.

Response: 

The authors would like to thank the reviewers about the comment. The motivation of investigating Ni-doped goethite NPs is mentioned in the second and third paragraphs from the end in the part of introduction (from line. 86 to 102).

Q2: Speaking about the results of X-ray diffraction, the authors note the absence of changes in the cell parameters for all concentrations, referring to the proximity of the radii of iron 3+ and nickel 2+. At the same time, according to X-ray absorption spectroscopy data, nickel is in an oxidation state greater than 2+. "According to the absorption edge's higher value compared to NiO, the oxidation state of Ni in all NixFe1-xOOH NPs samples is higher than +2." By the way, the question immediately arises: in what degree of oxidation more than 2+ can nickel be? Perhaps, to answer this question, it was necessary to measure standards not only with valence 0 and 2+, but also with valence 3+.

Response: 

We would like to thank the reviewer for their valuable comment. Similar question is raised by Reviewer1 regarding the oxidation state of Ni. Despite the known instability of Ni2O3, we intend to include it as a subject of investigation in our future work, aiming to achieve a more precise determination of Ni's oxidation state.

Q3: As for the hydrothermal synthesis of the compounds under study, it remains unclear from its description what causes the oxidation of nickel 2+ to nickel 3+ in a sealed hydrothermal cell during the synthesis of goethite. What was the oxidizing agent in this process?

Response:

We would like to thank the reviewer for their valuable comment. In the hydrothermal synthesis conditions we described in section 3.1 in the revised manuscript for the preparation of Ni-doped FeOOH, the oxidation of Ni(II) to Ni(III) is likely driven by a combination of factors:

Adding aqueous solution until the pH reaches 13 creates highly alkaline conditions. Under such conditions, the solution's hydroxide ions (OH-) can act as strong oxidizing agents. They can readily react with Ni(II) ions (Ni2+) to form Ni(III) species (Ni3+). The reaction can be represented as follows:

Ni2+ + 2OH-→ Ni(OH)â‚„3-

This reaction oxidizes Ni (II) from +2 to +3 oxidation state.

In addition, hydrothermal synthesis is carried out at an elevated temperature of 80°C. This temperature provides the necessary thermal energy for chemical reactions, including redox reactions. Also, using a Teflon-lined stainless-steel autoclave allows for the maintenance of high pressure inside the reaction vessel. This can enhance the solubility of oxygen in the aqueous solution and promote its availability as an oxidizing agent. Dissolved oxygen can react with Ni(II) ions, contributing to oxidation.

Q4: When determining the band gap, the authors used the extrapolation of the linear part in energy coordinates. There are no explanations in the text how they chose this linear part, and this choice looks very arbitrary. This creates doubts about the reliability of the obtained data on the dependence of the band gap on the nickel concentration.

Response:

We thank the reviewer for the comment. It is modified in the revised manuscript.

“For bandgap (Eg) estimation of powder samples, diffuse reflectance UV–Vis spectroscopy can be applied [40]. In this case, the detected light is reflected from the surface of the sample. Diffuse reflectance spectra (DRS) of α-NixFe1-xOOH NPs are shown in Figure. 10 (a). As for the DRS spectrum of Ni0, it can observe strong absorption at the wavelength of less than 380 nm, which relates to ligand-to-metal charge transitions [48]. A shoulder peak observed at 490 nm is attributed to two-step excitation, while weak absorption peaks of 610 and 890 nm are found due to the ligand field transition [48]. The relative absorption intensities of Ni5, Ni10, Ni15, Ni20, Ni30, Ni40, and Ni50 are larger than that of Ni0, indicating that the photon energy was absorbed by Ni-doped FeOOH NPs. Eg is an attribute of the photocatalyst, which establishes its wavelength active region. It can be revealed using transmission spectra in thin films or absorption for bulk samples or powder samples that can be dispersed in a solution.”

Figure 10 displays the plot of (hνα)² versus () obtained from diffuse reflectance spectroscopy (DRS) measurements of NixFe1-xOOH NPs, enabling estimation of the Eg values. The Eg value for pure goethite (Ni0) was determined to be 2.71±0.01 eV. Table 5 shows that the Eg values for all Ni-doped samples were smaller than that of Ni0. Notably, the smallest Eg value of 2.06±0.01 eV was observed for the Ni30 sample, determined as the x-axis intercept of the fitted straight lines depicted in Figure 10 (b). The Eg values of Ni40 and N50 were increased to 2.13±0.01 eV and 2.15±0.01 eV, respectively. The existence of Ni(OH)2 precipitated in Ni100x with x of larger than 30 was the reason for the increase of band gap energy because NiO6 in Ni(OH)2 has a similar environment to NiO.

Also, in Figure 10 (b), the dashed lines and arrows are drawn as guides for the eyes.

Q5: The photocatalytic properties of the synthesized compounds did not show a systematic behavior in a series of nickel concentrations. All samples show either the same properties as the original goethite, or even worse. The nickel10 sample alone showed an improvement, but there is no explanation in the text why this happened. By the way, it is not clear why the authors used hydrogen peroxide in the measurements. If it is a sacrificial reagent, then it would be necessary to give blank measurements without a catalyst in the presence of peroxide and in the presence of a catalyst without light.

Response:

We would like to thank the reviewer for the comments. We modified and included it in the revised manuscript.

Q6: In electrochemical experiments, no systematicity is observed either, and a sharp drop in the first cycles indicates the rapid degradation of all samples, which may be due to the presence of a large amount of adsorbed water and chemically bonded water. This is confirmed by TGA data. In this regard, the question arises as to the expediency of using such compounds as cathode materials. Again, the measurements performed by the authors of two-phase samples containing, according to XRD, nickel hydroxide, are surprising.

Response:

We express our gratitude to the reviewer for his valuable comment.  In the context of our forthcoming research endeavors, we have deliberated upon and successfully addressed this issue through a heat treatment procedure aimed at mitigating residual water content in our samples. Subsequently, we intend to assess the battery performance of the treated samples and juxtapose these results with the data presented in the current manuscript.

Q7: As for the conclusions, on the one hand they contain a lot of unnecessary information about not the most important results, on the other hand, there is no main conclusion about the role of the doped nickel for the studied properties of the synthesized compounds.

Response:

We would like to thank the reviewer for the comment. The conclusion was modified in the revised manuscript.

Round 2

Reviewer 1 Report

The authors have addressed all the comments and therefore, the revised manuscript can be considered for publication in IJMS.

Reviewer 2 Report

The authors' answers do not satisfy me very much. The authors express an erroneous opinion regarding the oxidizing effect of hydroxide ions, although I agree that dissolved oxygen can be an oxidizing agent for nickel 2+. In this case, it is difficult to expect reproducibility of the properties of the synthesized samples, since the authors do not control its quantity. I have no desire to continue the discussion, so the article can be accepted in this form.

No comments